# Automated model building and protein identification in cryo-EM maps

Kiarash Jamali[1✉], Lukas Käll[2], Rui Zhang[3], Alan Brown[4], Dari Kimanius[1✉] & Sjors H. W. Scheres[1✉]

Interpreting electron cryo-microscopy (cryo-EM) maps with atomic models requires high levels of expertise and labour-intensive manual intervention in three-dimensional computer graphics programs[1,2]. Here we present ModelAngelo, a machine-learning approach for automated atomic model building in cryo-EM maps. By combining information from the cryo-EM map with information from protein sequence and structure in a single graph neural network, ModelAngelo builds atomic models for proteins that are of similar quality to those generated by human experts. For nucleotides, ModelAngelo builds backbones with similar accuracy to those built by humans. By using its predicted amino acid probabilities for each residue in hidden Markov model sequence searches, ModelAngelo outperforms human experts in the identification of proteins with unknown sequences. ModelAngelo will therefore remove bottlenecks and increase objectivity in cryo-EM structure determination.

Knowledge of the three-dimensional atomic structures of proteins and nucleic acids is essential for our understanding of the molecular processes of life. In recent years, considerable advances have been made in the determination of structures of biological macromolecules using electron cryo-microscopy (cryo-EM), culminating in cryo-EM maps of proteins with sufficient resolution to resolve individual atoms[3,4]. Accordingly, the number of new cryo-EM structures in the Electron Microscopy Data Bank (EMDB)[5] is growing exponentially. If this trend continues, approximately 100,000 cryo-EM structures will be determined in the next 5 years[6].

Over two-thirds of the structures reported in 2022 had resolutions better than 4 Å. Although individual atoms are not resolved at resolutions between 2–4 Å, reliable atomic models can be built by exploiting previous knowledge of the chemical structures of the proteins and nucleic acids in the sample, including their amino acid and nucleic acid sequences. Typically, atomic model building in cryo-EM maps is performed using manual procedures in three-dimensional computer graphics programs[1,2]. Atomic model building is often time-consuming and requires substantial levels of expertise to produce accurate models. At resolutions better than 3 Å, experts can build atomic models with few errors, whereas, at resolutions below 4 Å, avoiding mistakes is challenging. It is therefore not uncommon for atomic models of biological complexes to contain errors[7], with potentially serious consequences[8].

Structure determination using cryo-EM is also an increasingly important tool for the discovery of new subunits in biological complexes. Owing to its relaxed requirements for sample quantity and purity compared with other structural biology techniques, cryo-EM can determine structures of complexes purified from endogenous sources. Many such complexes contain subunits of unknown identities. Without previous knowledge of the amino acid sequence, identifying the chemical identity of individual amino acids in cryo-EM maps is difficult, and requires relatively high resolutions. Yet, provided that one can build stretches of several consecutive amino acids, database searches with the sequence fragments can lead to the identification of the corresponding protein. Recent examples include the identification of TMEM106B in amyloid filaments from human brains[9–11] and the detection of subunits of axonemal complexes[12,13].

Here we introduce a machine-learning approach called ModelAngelo for the automated building of atomic models and the identification of proteins in cryo-EM maps. Machine-learning approaches often require large amounts of training data. For example, recent protein language models were trained on tens of millions of sequences[14] and AlphaFold2 was trained on more than 200,000 structures[15]. By contrast, fewer than 13,000 cryo-EM structures with resolutions better than 4 Å have been determined to date and many of these are redundant. The limited amount of available training data prompted us to design a multimodal machine-learning approach that combines local information from the cryo-EM map surrounding each protein or nucleic acid residue with additional information from the protein sequences in the sample and the local geometry of the structure. Similar sources of information are used by human experts when manually building atomic models in cryo-EM maps.

The sudden availability of atomic models for millions of proteins from protein structure prediction by AlphaFold2[15,16] has helped to guide and accelerate model building[17]. However, previous attempts to fully automate atomic modelling[18–24] or the identification of unknown proteins[25–27] have not become mainstream, although DeepTracer[21,24] and findMySequence[25] have gained some traction. However, atomic modelling remains a time-consuming and expert-dependent process in many structure determination projects. With the ongoing exponential growth in cryo-EM structures and the continuing influx of newcomers to the cryo-EM field, automation will be key in removing bottlenecks and replacing the dependence on human experts with objective methods that are accessible to all. We demonstrate that ModelAngelo can meet this need. Although subsequent error

[1]MRC Laboratory of Molecular Biology, Cambridge, UK. [2]Science for Life Laboratory, KTH Royal Institute of Technology, Stockholm, Sweden. [3]Washington University in St Louis, St Louis, MO, USA. [4]Blavatnik Institute, Harvard Medical School, Boston, MA, USA. ✉e-mail: kjamali@mrc-lmb.cam.ac.uk; dari@mrc-lmb.cam.ac.uk; scheres@mrc-lmb.cam.ac.uk

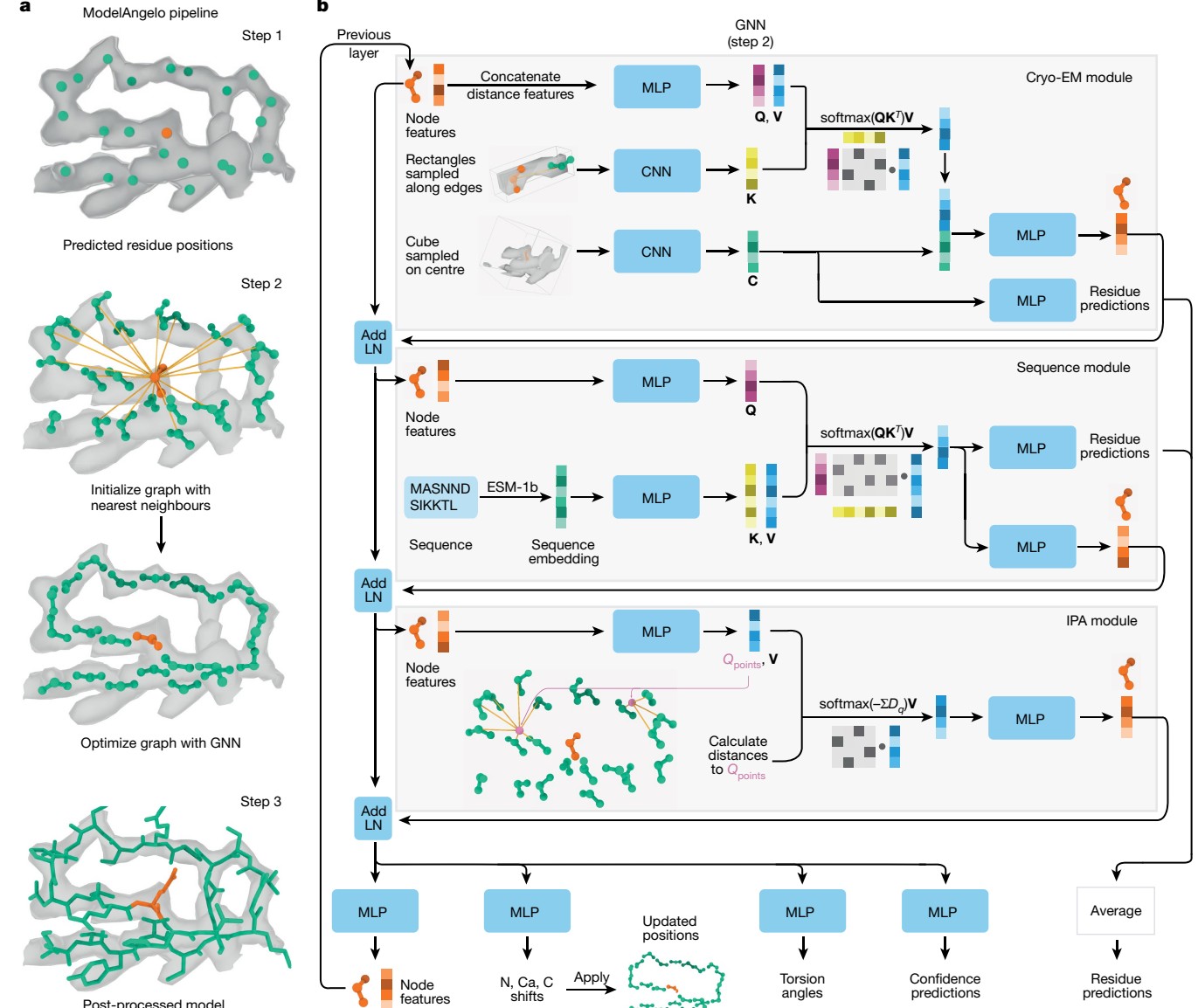

**Fig. 1 | Atomic modelling in ModelAngelo. a**, ModelAngelo builds atomic models in three steps: (1) a CNN predicts protein and nucleic acid residue positions; (2) a GNN optimizes these positions and orientations (shown in **b**); (3) post-processing of the optimized graph leads to a complete atomic model. **b**, The GNN, which is arranged in eight layers with three modules, uses a feature vector per residue that is passed through MLP and integrated with additional data through attention mechanisms that have query (**Q**), key (**K**) and value (**V**) vectors. The cryo-EM module also produces a feature vector (**C**) used for residue prediction. The IPA module uses query points ($Q_{points}$) and their distances to the neighbouring residues ($D_q$) for attention. Stable gradient propagation is ensured by residual connections with layer norms (Add LN)[51]. Residue feature vectors are used to update residue positions and orientations. They are also used to predict torsion angles, confidence scores and residue identities at the end of each layer.

checking and refinement remain necessary, ModelAngelo outperforms human experts in identifying unknown proteins and produces initial atomic models of comparable completeness to those obtained by human experts.

## A multimodal approach to model building

Automated model building of proteins and nucleic acids in ModelAngelo comprises three steps (Fig. 1a). Details about the network architectures that underlie these steps and how they are trained have been described previously[28].

In the first step, positions for the backbone Cα atom of amino acids and the phosphor atom of nucleic acids are predicted using a convolutional neural network (CNN). This CNN is a modified feature-pyramid network[29] that predicts whether each voxel in the cryo-EM map contains the Cα atom of an amino acid, the phosphor atom of a nucleic acid residue or neither. A graph is then constructed in which each residue is a node, and edges are formed between each residue and its 20 nearest neighbours.

In the second step, a graph neural network (GNN) is used to optimize the positions and orientations of the residues to predict their amino or nucleic acid identity, and to predict torsion angles for their side chains or bases. The GNN consists of three modules: a cryo-EM module, a sequence module and an invariant point attention (IPA) module (Fig. 1b). Each node of the graph is associated with a residue feature vector. Each module takes the residue feature vector as input, combines it with new information and outputs an updated residue feature vector that is passed to the next module. The sequential application of the

three modules in eight layers (Fig. 1b) enables the gradual extraction of more information from the different inputs.

The cryo-EM module incorporates information from the cryo-EM map and comprises two parts. First, the input feature vector is passed through a multilayer perceptron (MLP) network to generate query and value vectors. These vectors are used for cross-attention[30] with key vectors that are calculated from a CNN on rectangular boxes that are extracted from the cryo-EM density map that point from the current residue to its 20 nearest neighbours. Intuitively, the cross-attention mechanism allows mixing information from each residue with that of its 20 nearest neighbours, depending on whether the cryo-EM density between them looks connected. Second, a cubic box is extracted from the cryo-EM map around the position of the current residue and passed through another CNN. The resulting vector is used in two ways: to generate amino and nucleic acid identity predictions through an MLP; and, after concatenation with the vector from the cross-attention, it is passed through another MLP to generate the output residue feature vector of the cryo-EM module.

The sequence module performs cross-attention for each residue with the user-provided amino acid sequences, which are embedded using the pretrained protein language model ESM-1b[31]. This incorporates information that is learned by the language model from many amino acid sequences, including multiple homologues. The information in protein language models has been shown to be sufficient for protein structure prediction[14]. The vector from the cross-attention is used in two ways: a first MLP is used to generate amino and nucleic acid identity predictions; a second MLP generates the output residue feature vector of the sequence module. For nucleic acid residues, the sequence module is not used.

The IPA module incorporates information from the geometry of the nodes in the graph and was inspired by the module with the same name in AlphaFold2[15]. An MLP calculates four query points per residue and the Euclidean distance between the query points and the location of the neighbouring nodes is used to replace the cosine similarity of the attention algorithm between the query and key vectors. Intuitively, this enables the model to learn information about the topology of neighbouring residues, for example, about secondary structure. In fact, disabling this module in an ablation study led to atomic models with incorrect secondary structure geometry[28].

In the third and final step, the residue feature vectors are post-processed to generate an atomic model. The feature vectors are used as inputs into two separate MLPs to predict new positions and orientations for each residue, as well as torsion angles for amino acid side chains and nucleic acid bases. They are also used to predict a confidence score for each residue, which is based on the network's predicted root-mean-squared deviation (r.m.s.d.) for the backbone atoms with the deposited structure. Moreover, the predictions for the amino or nucleic acid identities from the cryo-EM and sequence modules are averaged to generate probabilities for each possible identity for all residues. These vectors are converted into a hidden Markov model (HMM) profile that is used for a search against the input sequences using HMMER[32]. A profile HMM is a probabilistic model representing the multiple-sequence alignment (MSA) of a set of related sequences. The parameters of a profile HMM are normally estimated from the MSA that it strives to model; however, here they are instead estimated from ModelAngelo predictions. There are three types of state in the profile HMM. For each position of the MSA's consensus sequence, there is a match (M), a delete (D) and an insert (I) state with respect to the query sequences[33]. There are two types of probabilities in a profile HMM: transition and emission. The transition probabilities reflect the probability of a sequence going between the M, I and D states from one position of the profile to the next. ModelAngelo uses the confidence metric, $c^{(i)}$, that it predicts for each residue $i$ to construct the transition probabilities as follows:

$$P_{M\to M}^{(i)} = \max(c^{(i)} - d, 0.5) \quad P_{D\to M}^{(i)} = 1 - d \quad P_{I\to M}^{(i)} = 1 - d$$

$$P_{M\to D}^{(i)} = \frac{1 - P_{M\to M}^{(i)}}{2} \qquad P_{D\to D}^{(i)} = d \qquad P_{I\to D}^{(i)} = 0$$

$$P_{M\to I}^{(i)} = \frac{1 - P_{M\to M}^{(i)}}{2} \qquad P_{D\to I}^{(i)} = 0 \qquad P_{I\to I}^{(i)} = d$$

The strategy to set $P_{M\to I}^{(i)} = P_{M\to D}^{(i)}$, the constant $d = 0.5$ and the minimum value of $P_{M\to M}^{(i)} = 0.5$ were chosen arbitrarily and these values were never optimized. The emission probabilities represent the probability of each amino acid being produced in an M or I state. For these, Model-Angelo uses its predicted probability distribution of the amino acids for each residue. The resulting HMM profiles are compatible with HMMER3[34] and HHblits[35]. Matched residues are mutated to the corresponding amino or nucleic acid in the input sequences, and separate chains are connected on the basis of their assigned sequences and proximity. Finally, chains shorter than four residues are pruned from the model, and a full atomic model is generated from the predicted positions and orientations of each residue and their corresponding amino acid or nucleic base torsion angle predictions using idealized geometries. The predicted backbone r.m.s.d. values are mapped to a score between 0 and 1, corresponding to a linear range for r.m.s.d. values between 1.2 and 0.5 Å, respectively. This score is stored in the B-factor column of the output coordinate file as a measure of local confidence in the backbone geometry.

Inspired by AlphaFold[15], we recycle the post-processed model from one round of the GNN as the starting point of a subsequent round of graph optimization. For this purpose, ModelAngelo was trained with a random number of 1–3 recycling steps. During inference, we perform three rounds of recycling, as the performance plateaus after three rounds.

We trained ModelAngelo on maps deposited in the EMDB[5] before 1 April 2022 with resolutions better than 4 Å and paired with models in the Protein Data Bank (PDB)[36] that cover the entire map correctly, as described previously[28]. PDB files that included insertion codes, that is, additional residues relative to the reference sequence, were removed. This resulted in 3,715 map–model pairs that were used during training. All cryo-EM maps were resampled to a common pixel size of 1 Å. For comparison, findMySequence uses only 117 pairs, while DeepTracer uses approximately 1,400 (refs. 21,25).

To enable model building for structures with unknown sequences, we also trained a version of ModelAngelo without its sequence module. Still, for each protein residue, ModelAngelo predicts probabilities for all 20 amino acids. Within ModelAngelo, these probabilities are converted into HMM profiles and used for searches in HMMER3[34] as described above, but using a larger proteome, rather than only the sequences known to be present in the structure.

## Protein modelling is on par with humans

To test ModelAngelo, we first considered all cryo-EM structures determined to at least 4 Å resolution and released from the EMDB between the cut-off date for training, 1 April 2022, and 9 February 2023. To reduce the computational costs, we excluded structures with more than 30,000 protein residues. We also removed viruses with icosahedral symmetry, for which typically only the asymmetric unit was built. To ensure that none of the sequences were seen before during training, we removed structures that had protein chains with more than 10% sequence identity to any of the proteins in the training set. Finally, we removed structures with insertion codes and other irregularities. This resulted in a test set of 177 structures (Supplementary Information), on which we ran ModelAngelo. Using a single A100 GPU, the smallest structure (PDB: 8DWI; molecular mass, 54.7 kDa) took 2 min; the largest structure (PDB: 7UMS; molecular mass, 1.85 MDa) took 53 min. The output coordinates from ModelAngelo were refined against the

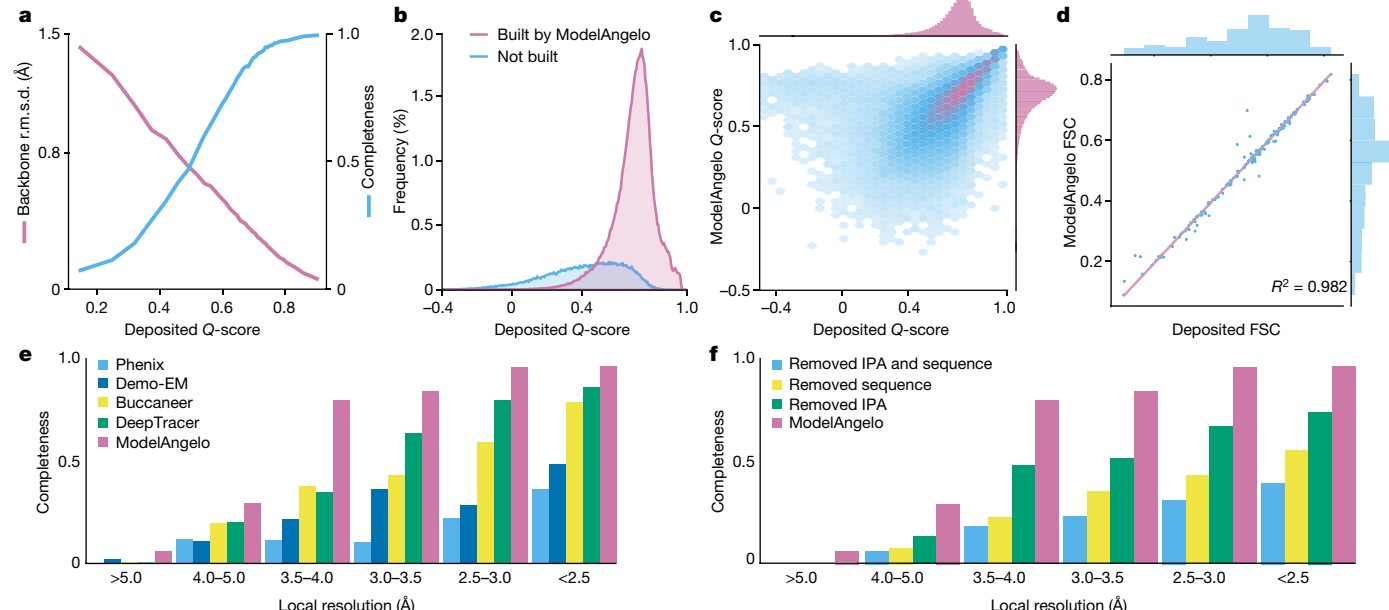

**Fig. 2 | Performance of ModelAngelo for proteins. a**, The backbone r.m.s.d. and model completeness plotted as a function of the target model $Q$-scores. **b**, Histograms of the $Q$-scores of residues in the deposited models, comparing those built by ModelAngelo with those not built. **c**, $Q$-score comparison between ModelAngelo-predicted models and the deposited models. **d**, Model-to-map Fourier shell correlation (FSC), as calculated by Servalcat[37] after refining both models and using only residues present in both ModelAngelo and deposited models. **e**, Model completeness for various automated model-building software for different local-resolution ranges in the maps. **f**, Model completeness for ModelAngelo and versions of ModelAngelo in which its sequence and/or IPA modules were ablated. For **a**–**d**, the data relate to the test set of 177 structures; for **e** and **f**, the data relate to the subset of 27 structures.

cryo-EM map using a standard refinement cycle in Servalcat[37], and the refined models were compared to the deposited ones.

To assess the quality of the models generated by ModelAngelo, we analysed the $Q$-scores[38] of all of the structures in the test set. The $Q$-score measures the resolvability of individual atoms in cryo-EM maps, therefore reflecting the quality of the built model. Provided that the model is built well, $Q$-scores also correlate with the local resolution, which can vary in cryo-EM maps: $Q$-scores of 0.4 are typical for cryo-EM maps at 4 Å resolution, values better than 0.7 are typical for maps beyond 2 Å resolution and values of 0.6 are typical for maps at 3 Å resolution[38]. We implemented $Q$-score calculation in ModelAngelo and calculated the average $Q$-scores for all atoms in each residue of both the deposited models and those built by ModelAngelo. We next calculated backbone r.m.s.d. values between the protein models built by ModelAngelo and those deposited and plotted these against the $Q$-scores of the deposited residues (Fig. 2a (pink line)). As expected, ModelAngelo builds models with lower r.m.s.d. values for residues with higher (better) $Q$-scores. Even for residues with $Q$-scores as low as 0.4, ModelAngelo builds models with backbone r.m.s.d. values lower than 1.0 Å. We also measured the completeness of the models built by ModelAngelo. We define completeness as the fraction of residues that are built with their Cα atom within 3 Å of the deposited model and with the correct amino acid assignment. As with backbone r.m.s.d., completeness improves for residues with higher $Q$-scores (Fig. 2a (blue line)). Overall, ModelAngelo built 77% of all 410,585 residues in the test set. Analysis of the deposited $Q$-scores shows that those residues not built by ModelAngelo have lower $Q$-scores than those that are built (Fig. 2b). In the deposited models, many of the residues with the lowest $Q$-scores were probably obtained by rigid-body docking of protein domains into poorly resolved regions of the cryo-EM maps. Excluding the 51,446 residues with $Q$-scores below 0.4, ModelAngelo built 85% of the residues in the test set. A comparison of $Q$-scores calculated for the models built by ModelAngelo with those calculated for the deposited models shows that models from ModelAngelo are of similar quality to the deposited ones (Fig. 2c).

The same is also true for overall Fourier shell correlation values between the cryo-EM maps and those parts of the models that were both built by ModelAngelo and present in the deposited models (Fig. 2d).

In a second test, we compared the performance of ModelAngelo with existing approaches for automated model building in cryo-EM maps. For this test, we used a subset of 27 protein structures from the 177 structures described above (Supplementary Information). We selected nine single-chain structures, nine homo-oligomeric structures and nine hetero-oligomeric structures. For each of these types of structures, we selected three structures with overall resolutions below 3.3 Å, three structures with resolutions between 3.3 and 2.8 Å, and three structures with resolutions better than 2.8 Å. For all 27 structures, unfiltered half-maps were available for download from the EMDB, and we used these to calculate local resolutions in ResMap[39]. We then used Phenix[40], Demo-EM[41], Buccaneer[20] and DeepTracer[21] for automated model building in these maps and compared the completeness of the resulting models with those obtained using ModelAngelo (Fig. 2e and Extended Data Table 1). The best alternative approach, DeepTracer, built approximately 80% of the deposited residues in regions of the maps with local resolutions in the range of 2.5–3 Å; the remaining approaches built models with considerably lower completeness. By contrast, ModelAngelo built up to 80% of the deposited residues in regions of the maps with local resolutions down to 3.5–4 Å, reflecting the observation that manual building by human experts also becomes prone to errors at resolutions below 4 Å. Tests in which we ran ModelAngelo without one or more of its modules indicate that its performance comes from a combination of all three modules (Fig. 2f), which is consistent with previous observations[28].

## Building good nucleic acid backbones

The test set of 177 structures described above contained only 103 nucleic acid chains, many with just a few nucleotides. Thus, instead of conducting a systematic analysis as done for the proteins, we present

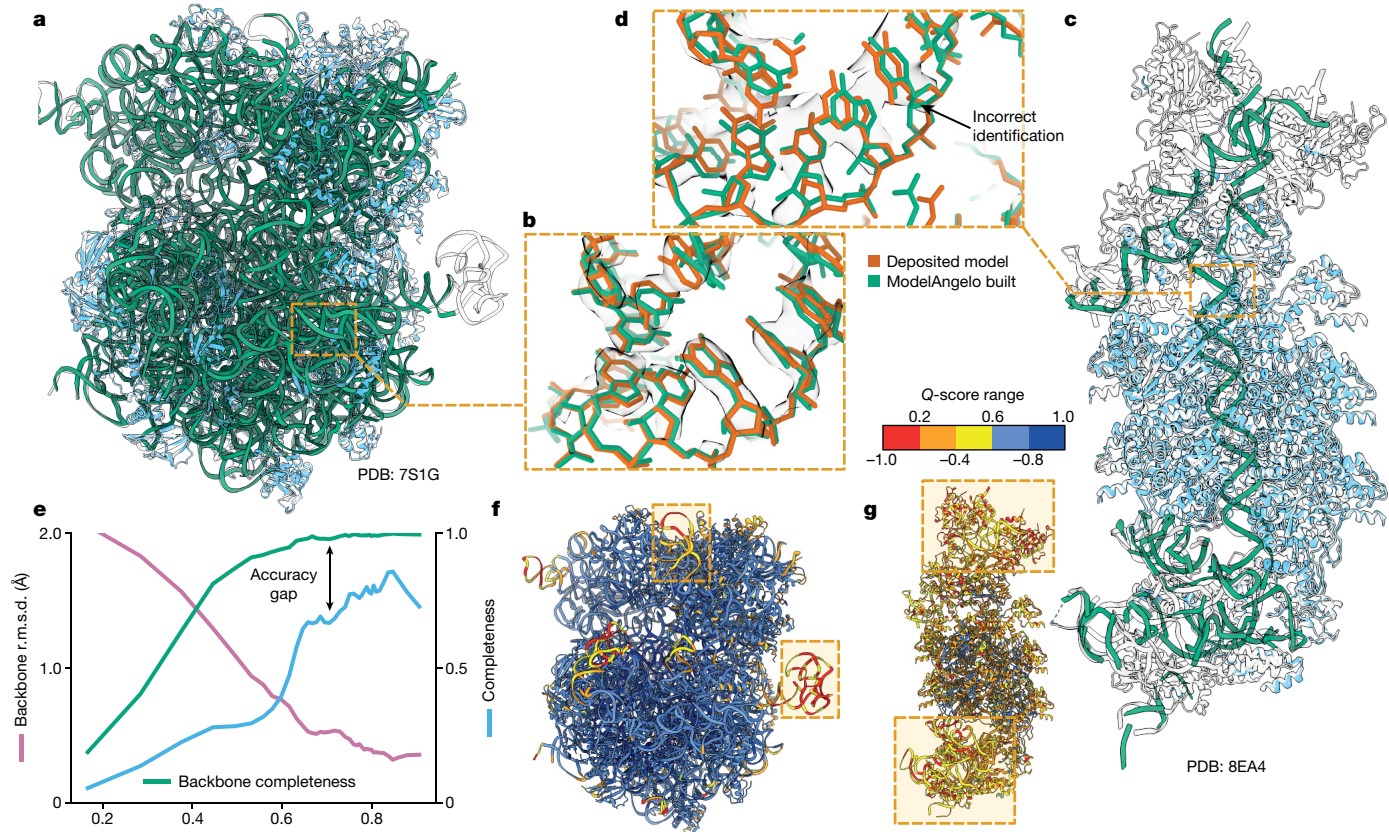

**Fig. 3 | Performance of ModelAngelo for nucleic acids. a**, *Escherichia coli* ribosome built by ModelAngelo (with ribosomal RNA in green and proteins in blue) compared with the deposited model (PDB: 7S1G, black outline)[52]. **b**, Magnified view with nucleotide bases showing high accuracy compared with the deposited model (orange). **c**, ModelAngelo model of the V-K CAST transpososome from *S. hofmanni* compared with the deposited model (PDB: 8EA4)[42]. Sections that were not built by ModelAngelo (black outline) are in regions of low *Q*-score (as shown in **g**). **d**, Magnified view comparing the nucleotide bases of both models, showing a sequence that was incorrectly identified by ModelAngelo. **e**, Backbone r.m.s.d., backbone completeness and sequence completeness were plotted against the deposited *Q*-score for six ribosome structures. **f,g**, Deposited models for the structures in **a** and **c**, respectively, coloured by *Q*-score, with low-*Q*-score regions indicated by boxes.

a few test cases to illustrate the quality of nucleotide building (Fig. 3). We applied ModelAngelo to 11 different ribosome structures that were determined to resolutions ranging from 1.98 to 3.80 Å (Fig. 3a,b), as well as a CRISPR-associated transpososome from *Scytonema hofmanni*[42] (Fig. 3c,d). Although ribosome structures were included in Model-Angelo's training set, the nucleotide sequences were not. When plotting backbone r.m.s.d. values and backbone completeness against the *Q*-scores of the deposited nucleotide coordinates (Fig. 3e), we observed similar trends to those for the protein chains. Backbone r.m.s.d. values range from 2 Å in the worst regions of the map to values better than 0.5 Å in the best regions. Likewise, near-complete backbones are built in the best regions, while backbone completeness drops to below 80% for the worst regions. However, ModelAngelo struggles to distinguish between the two purines or the two pyrimidines, echoing the difficulty that humans face in building nucleotide sequences based solely on the cryo-EM density, if the resolution does not extend beyond 2.5 Å. Consequently, when considering only correctly built sequences, the completeness of the models built by ModelAngelo drops to 80% for the best parts of the map, and to as low as 20% for the worst parts (Fig. 3e). Users should therefore carefully validate the nucleotide chains of models built by ModelAngelo, for example, by using nucleotide secondary structure predictors[43]. Nonetheless, ModelAngelo considerably accelerates the process of building the nucleotide backbone, as subsequent nucleotide base changes can be made with minimal manual intervention. For the CRISPR-associated transpososome and 3 out of the 11 ribosomes described above, we also used DeepTracer[26]

and CryoREAD[44]. ModelAngelo produced nucleotide models that were more complete and more accurate than these alternative approaches (Extended Data Table 2).

## Identifying novel proteins

To illustrate the performance of ModelAngelo in identifying protein chains in cryo-EM maps, we applied ModelAngelo to two examples of large cryo-EM structures that were recently determined from endogenous sources. The first example is a structure of the supercomplex of the phycobilisome (PBS), photosystem I and II (PSI and PSII) and the transmembrane light-harvesting complexes (LHCs) that was imaged in situ in the red alga *Porphyridium purpureum*[45]. The second example is a structure of the ciliary central apparatus and radial spokes of the green alga *Chlamydomonas reinhardtii* that was obtained by single-particle analysis after purification from cilia[12,13].

At 16.7 MDa, the PBS–PSII–PSI–LHC supercomplex is one of the largest complexes determined using single-particle cryo-EM. The deposited model (PDB: 7Y5E) consists of 158,730 residues in 81 unique protein chains, including six chains for which the authors were unable to identify the corresponding protein. The unidentified chains were termed LPP1 (linker of PBS–PSII 1); CNT (for connector); PsbW and Psb34 (two of the core subunits of PSII); LRH (a linker protein); and LPS1 (photosystem linker protein 1). To identify these chains, we ran ModelAngelo without using its sequence module (using the build_no_seq option) to calculate an initial atomic model with HMM profiles for all chains,

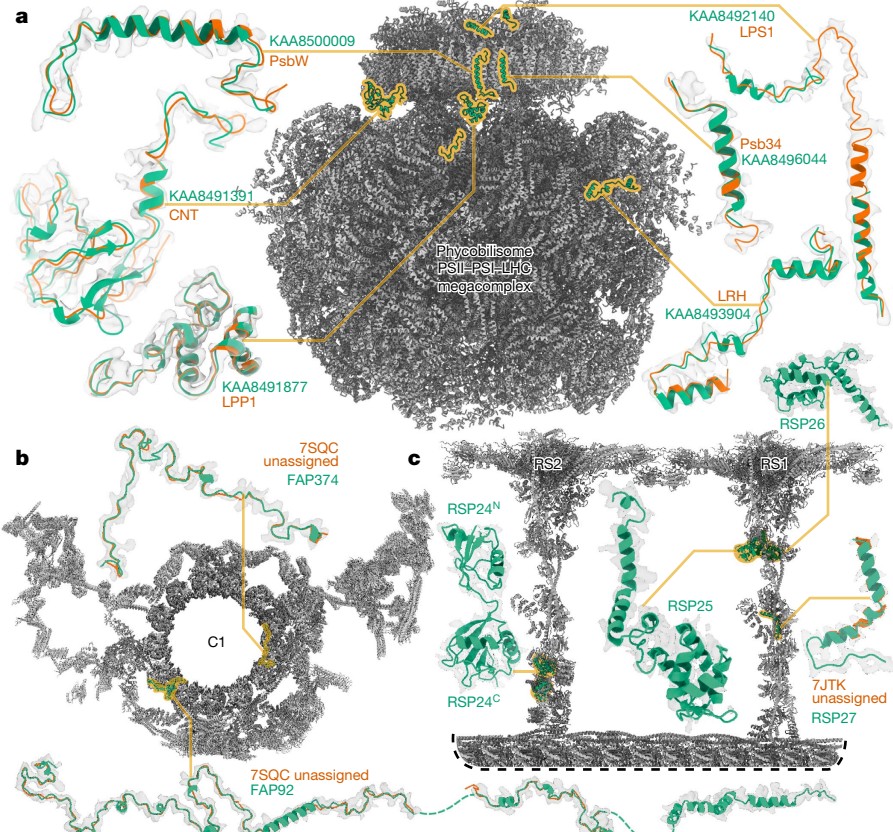

**Fig. 4 | Examples of protein identification using ModelAngelo. a,** The ModelAngelo model of the single-PBS–PSII–PSI–LHC supercomplex (grey) showing the positions, models and map densities of six newly identified proteins (green). Backbone traces in the deposited model (PDB: 7Y5E) are shown in orange. **b,** Atomic model of the central apparatus microtubule C1 showing the positions, models and map densities of two identified proteins— FAP92 and FAP374. The orange cartoons represent poly(UNK) chains deposited in the original model (PDB: 7SQC). **c,** An atomic model of radial spokes 1 and 2 (RS1 and RS2) bound to a doublet microtubule (grey) showing the positions, models and map densities of four proteins (RSP24–27, green) identified by ModelAngelo. Only RSP27 had a backbone trace in the deposited model (orange). C, C terminus; N, N terminus.

and we searched these profiles against the proteome constructed in ref. 46 (using the hmm_search option). Due to local pseudosymmetry, all six unidentified proteins occur more than once in the cryo-EM map. This enables us to bootstrap weaker individual hits by cross-referencing their matches to the other instances. Specifically, the same six protein chains were identified for all instances, with $E$-values in the range of $5.8 \times 10^{-66}$ to $6.4 \times 10^{-2}$. Using the backbone traces in the deposited model, find-MySequence[25] identified only two of the unassigned proteins (Psb34 and PsbW). Using the backbone traces generated by ModelAngelo, it also found LRH. We next constructed an input sequence file that included all chains in the deposited model plus the six newly identified chains and ran ModelAngelo again. This calculation took 23 h on an A100 GPU. The resulting model, containing 110,742 residues, is shown in Fig. 4a. For most sections of the unidentified chains, ModelAngelo built better models than those in the deposited structure, most notably for LRH and CNT. ModelAngelo did not build models for parts of the unidentified proteins that were in regions of poor cryo-EM density. Besides the excellent agreement between side-chain densities in the cryo-EM map and the predicted sequences (Extended Data Fig. 1), the structures built by ModelAngelo were also highly similar to AlphaFold2 predictions for the unidentified chains[15,47] (Extended Data Fig. 2). Mod-elAngelo did not attempt to build amino acid or nucleotide residues in the densities for phycocyanobilin or phycoerythrobilin cofactors (Extended Data Fig. 3). As the cryo-EM maps that ModelAngelo was trained on did contain cofactor densities, but it was trained to build protein and nucleic acid residues, ModelAngelo has been incentivized to ignore cofactor densities.

Like the PBS–PSII–PSI–LHC supercomplex, the central apparatus and radial spoke complexes isolated from *C. reinhardtii* ciliary axonemes are large complexes with poorly characterized subunit compositions. Although recent cryo-EM structures had identified 23 different radial spoke proteins (RSPs) and 48 different central apparatus proteins[12,13], the deposited maps (EMDB: EMD-22475, EMD-24481 and EMD-25381) contained densities that were left unassigned despite considerable manual effort. To identify these proteins, we applied ModelAngelo without using its sequence module to the deposited maps and searched the resulting HMM profiles against the latest version of the *C. reinhardtii* predicted proteome[48] (Fig. 4b and Methods). This approach identified four additional radial spoke proteins: FAP109, Cre05.g240450, Cre08. g800895 and Cre17.g802036), which we rename RSP24, RSP25, RSP26 and RSP27, respectively, and two additional central apparatus proteins (FAP92 and FAP374) (Extended Data Table 3). Using ModelAngelo's backbone traces, findMySequence[25] was unable to identify any of these proteins. Neither RSP24 nor RSP26 were annotated in earlier versions of the *C. reinhardtii* genome, explaining their absence from proteomic studies, and demonstrating the importance of high-quality genome annotations for de novo identification of proteins by cryo-EM. RSP27 was identified from a fragment of just 33 residues, demonstrating the power of ModelAngelo to identify proteins from small sections of well-resolved density. Both central apparatus proteins (FAP92 and FAP374) bind directly to the microtubule surface and have tertiary structures that are poorly predicted by AlphaFold2 (Extended Data Fig. 4); side-chain density was therefore essential for their success-ful identification (Extended Data Fig. 5). The identification of these

proteins will allow their functional relevance to the regulation of ciliary motility to be investigated through targeted genetic manipulation.

## Discussion

ModelAngelo automates atomic modelling in cryo-EM maps, building protein models of comparable quality to those built by human experts and nucleic acid models with near-complete and accurate backbones. ModelAngelo outperforms existing approaches for the automated modelling of both proteins and nucleotides. Furthermore, Model-Angelo builds these models within hours on a modern GPU, thereby removing an important bottleneck in cryo-EM structure determination. Future incorporation of ModelAngelo into automated cryo-EM image-processing pipelines will enable users to go from data acquisition to atomic models in a single automated procedure.

By introducing objectivity in the model-building process, ModelAngelo also informs which parts of the map can be confidently interpreted with an atomic model and which should be left uninterpreted. In this way, ModelAngelo will not only reduce the number of errors in atomic models but also have a role in making cryo-EM structure determination more accessible to the large numbers of newcomers that the field has experienced in recent years. Still, some degree of human supervision and intervention will remain necessary. Models from ModelAngelo will still need refinement, for example, in Servalcat[37] or Phenix[40], to optimize their stereochemistry and fit to the cryo-EM map. Users are also strongly encouraged to manually check the output of ModelAngelo, particularly for those parts of cryo-EM maps with resolutions worse than 3.5–4.0 Å, as rigid-body fitting of known domains or connecting loops in lower-resolution map regions to obtain a more complete model falls outside the scope of ModelAngelo. Colouring the model by its predicted confidence in backbone geometry, as stored in the *B*-factor column of the coordinate file, may guide the user towards parts of the model that are less reliable. ModelAngelo was trained with augmentation through a variety of positive and negative *B*-factors. It should therefore be relatively stable to local variations in *B*-factor. It is possible that combining ModelAngelo with neural networks that make cryo-EM maps look more like proteins[49,50] could lead to further improvements, although this would probably require retraining of ModelAngelo to reach its full potential.

Besides accelerating cryo-EM structure determination and providing objectivity in atomic modelling, ModelAngelo also identifies protein chains in cryo-EM maps better than human experts. The reason why ModelAngelo outperforms the human expert in this task probably lies in the implementation of its sequence searches. While human experts typically base their identifications on discrete assignments of individual amino acids to various residues in unknown chains, ModelAngelo exploits predicted probabilities for all 20 amino acids for every protein residue and combines this information with its predicted confidence in each residue in a full HMM search. This not only allows better identification of unknown chains but also helps ModelAngelo during the building of atomic models with known sequences, where it may potentially outperform human experts in placing protein chains for which ambiguity exists, for example, when multiple homologous chains coexist in a single structure. The ability to identify proteins in cryo-EM maps will increase in importance as ongoing advances in sample preparation, microscopy and image processing enable ever more structures to be determined for samples purified from native sources or visualized in situ by electron tomography of frozen cells or thin tissue sections.

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

## Methods

### Changes in ModelAngelo 1.0

We previously described an early (beta) version of ModelAngelo[28]. Here we introduce the first stable release of ModelAngelo (v.1.0), which extends the beta version by adding the ability to build nucleotides and an updated HMM algorithm, as described in the main text. We also made minor changes in the GNN to improve the performance of ModelAngelo due to the enhanced requirements of building nucleotides. Whereas the beta version of ModelAngelo used cryo-EM maps to a maximum spatial frequency of 3 Å, ModelAngelo v.1.0 uses information up to 2 Å resolution. To capture the same context radius, the regions that are sampled around each residue in the cryo-EM module were therefore increased from 17 to 23 voxels for the cubes and from 5 to 7 voxels for the rectangle lengths. We improved the training of the model by using the Lion optimizer[53] and changing the dropout probability to 0.1 from 0 (ref. 54). To compensate for the increased computational costs of these changes, we also implemented several approaches to speed up calculations. In particular, ModelAngelo can now be run using multiple GPUs simultaneously, node updates are performed more efficiently and we use larger batch sizes in training. Furthermore, we confirmed that half-precision inference (running the model with a two-byte floating-point precision rather than the default four-byte one) does not affect the outcome in the GNN. As a result of these changes, ModelAngelo 1.0 runs faster than the beta version, even though it uses a larger network.

### Radial spoke and central apparatus

The structure of radial spoke 1 (RS1) from *C. reinhardtii* (EMD-22475)[12] contained unassigned proteins that were either left unmodelled or tentatively interpreted with a poly(UNK) model. To identify these proteins, we ran ModelAngelo without using its sequence module to calculate an initial atomic model with HMM profiles for all chains. We subsequently searched the HMM profiles against the latest version of the *C. reinhardtii* genome[48], which was not available at the time of the original publication. For a known radial spoke protein, RSP6, ModelAngelo correctly predicted 67% of all residues even without knowledge of its sequence. This approach also unambiguously identified three unassigned proteins: FAP109, Cre17.g802036 and Cre05.g240450, which we reassign as RSP25, RSP26 and RSP27, respectively. RSP27 was identified from a fragment of just 33 residues, demonstrating ModelAngelo's ability to identify proteins from minimal information, given well-resolved side-chain densities.

RSP25 and RSP26 form a heterodimer in the neck of RS1. These structurally similar proteins each have an N-terminal RIIa domain (similar to the dimerization-anchoring domain of cAMP-dependent protein kinase regulatory subunit) followed by two C-terminal EF-hand motifs. The proteins were identified on the basis of sequence differences between their better-resolved RIIa domains, demonstrating ModelAngelo's ability to distinguish between similar proteins. RSP25 (FAP109) had been detected by mass spectrometry analysis of RS1 purified from *C. reinhardtii* axonemes[12], providing confidence to the assignment. RSP26 (Cre17.g802036) was not annotated in earlier versions of the *C. reinhardtii* genome, explaining its absence from proteomic studies. RSP27 (Cre05.g240450) forms a small, L-shaped helix in the centre of the *RS1* stalk.

After identification, we constructed an input sequence file that included all of the chains in the deposited model along with the three newly identified chains and ran ModelAngelo again. This approach identified and built extensions of RSP16 that had been left unassigned in the deposited model. We then extended the models of RSP25 and RSP26 using AlphaFold2 predictions for the EF-hand motifs, which have relatively poor cryo-EM density, demonstrating how ModelAngelo and AI-based structure prediction methods can be used together to build more complete atomic models.

The microtubule-bound stalk of radial spoke 2 (RS2), which is structurally and compositionally different from RS1, also contained unassigned proteins in the deposited map (EMD-22481)[12]. We therefore applied the same process to identify one additional protein, Cre08.g800895, which we rename RSP24. RSP24 is a 25 kDa bilobal protein with an N-terminal ubiquitin-like domain. An LC8-interacting protein in the stalk of RS2 remains unassigned due to too few resolved side chains.

In the axoneme, radial spokes interact transiently with the central apparatus. Structures of the two microtubules (C1 and C2) that together form the *C. reinhardtii* central apparatus have recently become available (EMD-25381 and EMD-25361)[13]. The map of the C1 microtubule (EMD-25381) contained a number of unassigned densities. We therefore applied ModelAngelo without using its sequence module to a Phenix auto-sharpened version of the map, as the original map was post-processed using DeepEMhancer[49]. This approach identified two new proteins: FAP92 and FAP374. FAP92 is a microtubule-associated protein that binds in the interprotofilament cleft between protofilaments 3 and 4 and repeats with 32 nm periodicity, whereas FAP374 is a microtubule inner protein that repeats with 16 nm periodicity. Neither protein has a globular fold nor is fully resolved in the map, demonstrating ModelAngelo's ability to identify ordered fragments of proteins. The final models of FAP92 and FAP374 were extended manually using Coot[1] through regions of less-well-resolved density and refined in Phenix[55].

### Reporting summary

Further information on research design is available in the Nature Portfolio Reporting Summary linked to this article.

## Data availability

All atomic models described in this paper and built by ModelAngelo are available for download as a single archive from Figshare (https://doi.org/10.6084/m9.figshare.25218434).

## Code availability

ModelAngelo is freely available online under the open-source MIT license (https://github.com/3dem/model-angelo).

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

**Acknowledgements** We thank G. Ghanim, J. Greener, K. Naydenova, J. Schwab, Z. Sekne, S. Lövestam and K. Yamashita for discussions; M. Gui for contributions to atomic modelling of the ciliary axonemes; and J. Grimmett, T. Darling and I. Clayson for help with high-performance computing. This work was supported by the Medical Research Council as part of the United Kingdom Research and Innovation (MC_UP_A025_1013 to S.H.W.S.); the EU Horizon 2020 research and innovation programme (under grant agreement no. 895412 to D.K.); the National Institutes of Health (R01-GM141109 to A.B. and R01-GM138854 to R.Z.); and the Knut and Alice Wallenberg Foundation (2022.0032 to L.K.). For the purpose of open access, the MRC Laboratory of Molecular Biology has applied a CC BY public copyright license to any author accepted manuscript version arising.

**Author contributions** K.J. designed and implemented ModelAngelo. L.K. designed and implemented the HMM search algorithm. R.Z. and A.B. analysed ciliary axoneme data. D.K. and S.H.W.S. jointly supervised the project. All of the authors contributed to the writing of the manuscript.

**Competing interests** The authors declare no competing interests.

**Additional information**
**Correspondence and requests for materials** should be addressed to Kiarash Jamali, Dari Kimanius or Sjors H. W. Scheres.

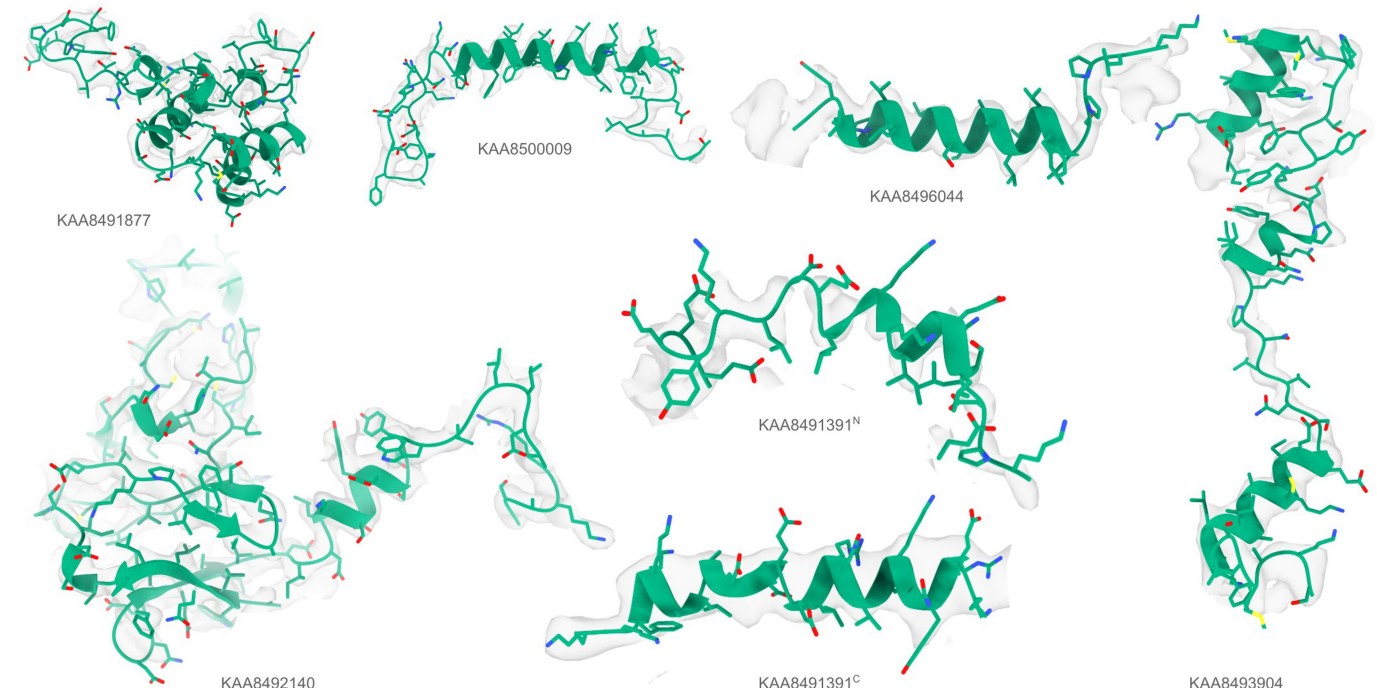

**Extended Data Fig. 1 | Identified proteins in the phycobilisome.** Atomic models built by ModelAngelo (green) for the six proteins that were identified by ModelAngelo. Side chain densities in the cryo-EM map (transparent grey) are in agreement with those of the atomic models.

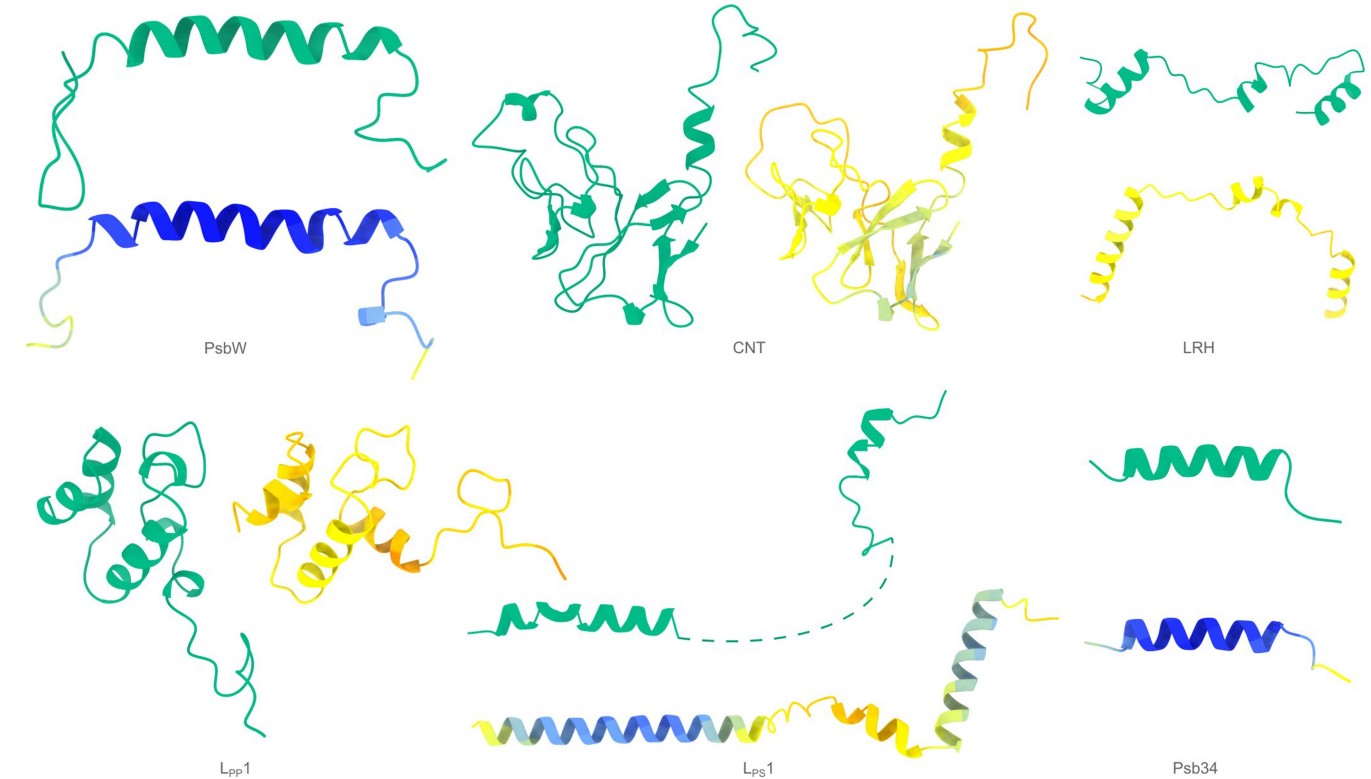

**Extended Data Fig. 2 | Models by ModelAngelo and AlphaFold for identified proteins in the phycobilisome.** Models built by ModelAngelo (green) are shown next to predictions of the corresponding sequences by AlphaFold (coloured by AlphaFold's confidence from high in blue, to low in red).

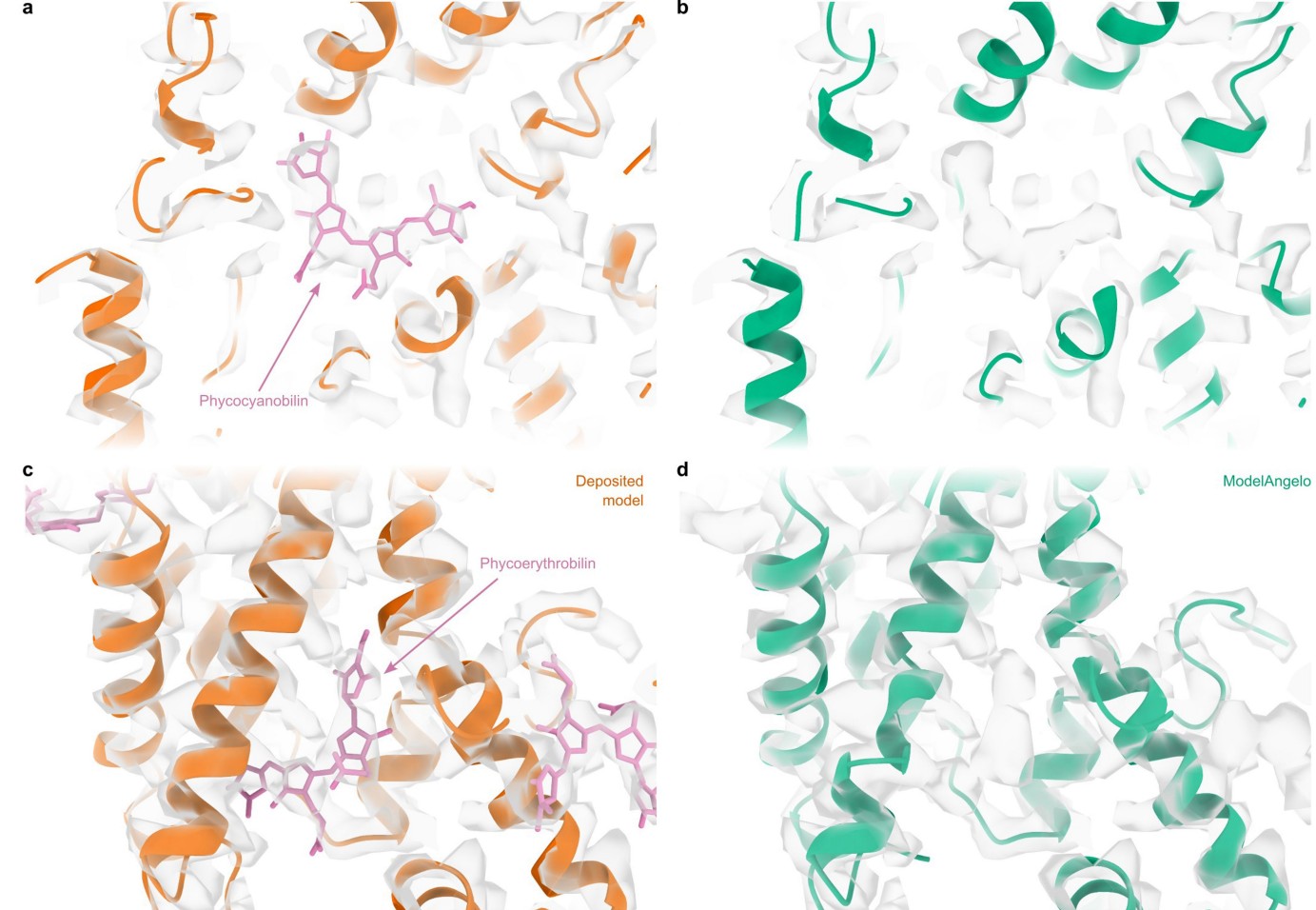

**Extended Data Fig. 3 | Performance around cofactors in the phycobilisome.** **a**, Cartoon representation of protein backbones (orange) and stick representation of a phycocyanobilin co-factor (pink) in the cryo-EM density (transparent grey) for the deposited phycobilisome structure. **b**, as in panel **a**, but for the model built by ModelAngelo (green). ModelAngelo leaves the cofactor density empty. **c**, **d**, as in panels **a**, **b** but for a phycoerythrobilin cofactor.

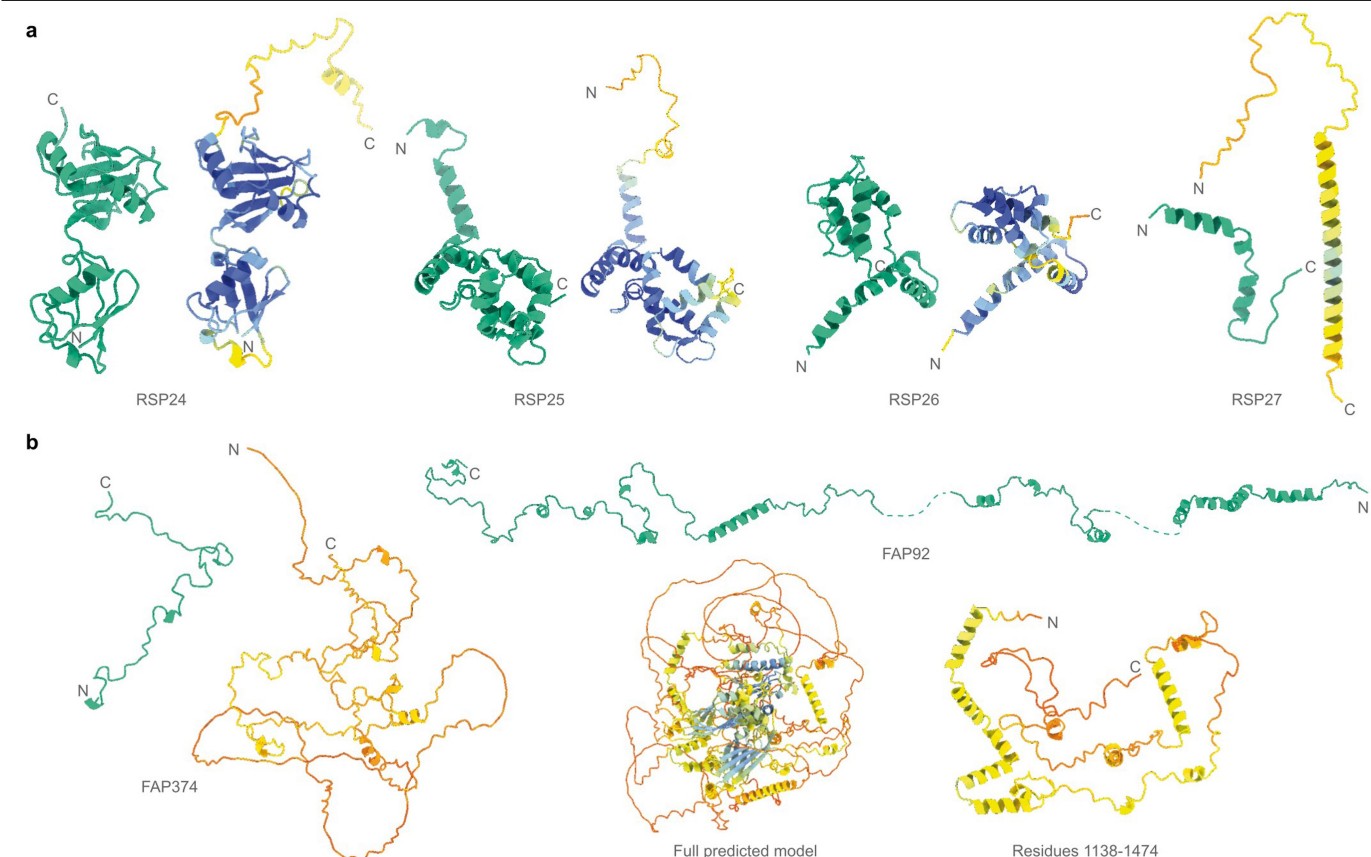

**Extended Data Fig. 4 | Models by ModelAngelo and AlphaFold for identified proteins in the ciliary axoneme.** Models built by ModelAngelo (green) are shown next to predictions of the corresponding sequences by AlphaFold (coloured by AlphaFold's confidence from high in blue, to low in red). These are split between **a**, the radial spoke proteins, and **b**, the central apparatus microtubule proteins.

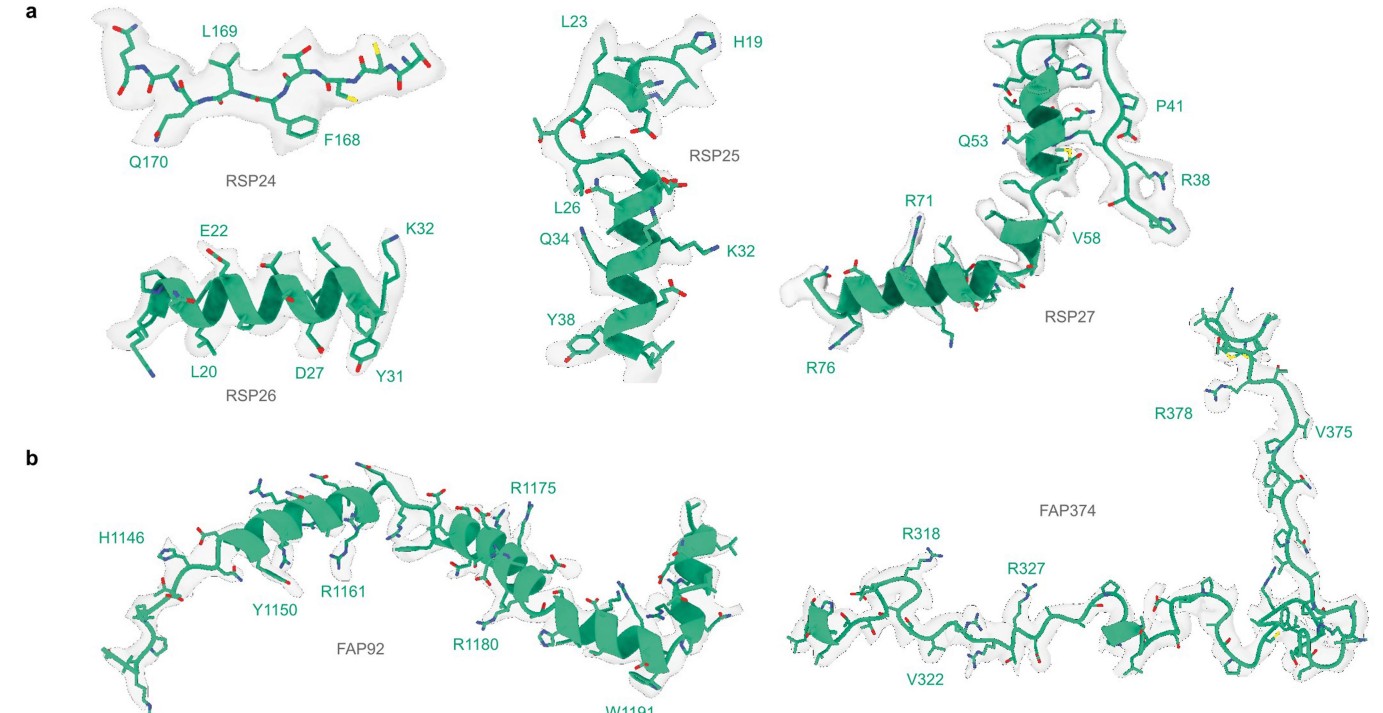

**Extended Data Fig. 5 | Identified proteins in the ciliary axoneme.** Atomic models built by ModelAngelo (green) for the six proteins that were identified by ModelAngelo. Side chain densities in the cryo-EM map (transparent grey) are in agreement with those of the atomic models. These are split between **a**, the radial spoke proteins, and **b**, the central apparatus microtubule proteins.

**Extended Data Table 1 | Comparison with alternative approaches for the automated building of proteins**

| PDB | Resolution (Å) | Backbone RMSD (Å) DT | MA | Calpha RMSD (Å) DT | MA | Backbone recall DT | MA | Backbone precision DT | MA | Accuracy DT | MA | Completion DT | MA |
|---|---|---|---|---|---|---|---|---|---|---|---|---|---|
| 7uzs | 2.2 | 0.514 | 0.107 | 0.372 | 0.09 | 96.7 | 91.9 | 70.6 | 96.3 | 98.9 | 99.8 | **95.6** | 91.8 |
| 7v0q | 2.5 | 0.644 | 0.105 | 0.394 | 0.09 | 95.6 | 98.8 | 54.4 | 97.6 | 92.8 | 99.8 | 88.8 | **98.6** |
| 7xgr | 2.6 | 0.827 | 0.33 | 0.516 | 0.303 | 94 | 92.9 | 63.6 | 68.3 | 86.2 | 99.2 | 81 | **92.1** |
| 7xjp | 2.71 | 1.127 | 0.333 | 0.591 | 0.311 | 94.5 | 92.6 | 97.4 | 99.7 | 59.5 | 98.5 | 56.2 | **91.2** |
| 7xk4 | 3.1 | 0.707 | 0.226 | 0.497 | 0.194 | 97.1 | 95.8 | 98.3 | 99.7 | 86.4 | 99.4 | 83.9 | **95.3** |
| 7xmv | 2.6 | 0.504 | 0.152 | 0.346 | 0.123 | 97.8 | 99.3 | 32.9 | 28.7 | 97.4 | 99.9 | 95.3 | **99.2** |
| 7xnz | 3.6 | 0.847 | 0.42 | 0.653 | 0.366 | 96.1 | 97.2 | 98.2 | 99.4 | 88.7 | 96.9 | 85.2 | **94.2** |
| 7yim | 2.6 | 1.483 | 0.448 | 0.91 | 0.407 | 69.9 | 49.8 | 91 | 98.3 | 14.5 | 92.2 | 10.1 | **45.9** |
| 7ypx | 3.12 | 1.212 | 0.674 | 0.844 | 0.587 | 57.6 | 73.1 | 98.2 | 97.1 | 55.9 | 95 | 32.2 | **69.5** |
| 7zh0 | 3.2 | 1.362 | 0.51 | 0.779 | 0.454 | 92.9 | 80.4 | 93.9 | 98 | 65.9 | 95.2 | 61.2 | **76.6** |
| 7zh6 | 3.67 | 1.405 | 0.581 | 1.096 | 0.548 | 90.6 | 67.8 | 87.8 | 98.5 | 41.1 | 95.4 | 37.2 | **64.6** |
| 8a04 | 3.2 | 0.84 | 0.246 | 0.453 | 0.196 | 95 | 100 | 10 | 12.2 | 92.2 | 100 | 87.6 | **100** |
| 8a7d | 3.06 | 1.085 | 0.42 | 0.729 | 0.36 | 82.9 | 79 | 96.2 | 99.3 | 71.6 | 97.3 | 59.4 | **76.9** |
| 8ap7 | 2.7 | 0.657 | 0.116 | 0.333 | 0.102 | 98.2 | 97.3 | 84 | 90.9 | 97 | 99.9 | 95.2 | **97.3** |
| 8ap8 | 3.7 | 0.861 | 0.356 | 0.579 | 0.314 | 97.1 | 94.9 | 93.9 | 98.6 | 79.3 | 98.6 | 77.1 | **93.6** |
| 8avx | 3.5 | 2.025 | 0.753 | 1.396 | 0.684 | 81.1 | 19.8 | 65.4 | 99.5 | 9.4 | 82.6 | 7.6 | **16.3** |
| 8bc2 | 2.6 | 0.5 | 0.134 | 0.35 | 0.127 | 99.6 | 100 | 98.2 | 100 | 98.9 | 100 | 98.5 | **100** |
| 8csw | 2.5 | 0.534 | 0.113 | 0.387 | 0.097 | 95.4 | 99.1 | 67 | 96.9 | 93.1 | 100 | 88.9 | **99.1** |
| 8cvz | 3.52 | 1.314 | 0.435 | 0.802 | 0.375 | 86.6 | 77.9 | 96.1 | 99.1 | 54.6 | 96.4 | 47.3 | **75.2** |
| 8dh7 | 2.99 | 0.664 | 0.188 | 0.475 | 0.169 | 98.7 | 99.3 | 97.6 | 100 | 86.6 | 99.7 | 85.5 | **99** |
| 8dnm | 2.76 | 0.743 | 0.175 | 0.438 | 0.143 | 99.3 | 99.6 | 99.7 | 99.9 | 96.7 | 99.9 | 96 | **99.6** |
| 8dwi | 3.4 | 1.111 | 0.612 | 0.827 | 0.571 | 96.4 | 95.9 | 59.7 | 96.4 | 56.1 | 94 | 54.1 | **90.1** |
| 8dwu | 3.4 | 1.727 | 0.622 | 0.952 | 0.558 | 39.4 | 34.8 | 97.5 | 98.9 | 45.2 | 94 | 17.8 | **32.7** |
| 8e50 | 3.67 | 1.212 | 0.397 | 0.801 | 0.339 | 96.5 | 92.8 | 78.9 | 99.7 | 49 | 98.3 | 47.3 | **91.2** |
| 8efe | 3.8 | 1.573 | 0.77 | 0.966 | 0.682 | 66 | 33.9 | 94.8 | 98.7 | 21.5 | 91 | 14.2 | **30.8** |
| 8evu | 2.58 | 0.836 | 0.122 | 0.537 | 0.105 | 98.1 | 99.7 | 92.5 | 99.4 | 88.5 | 100 | 86.8 | **99.7** |
| 8fma | 3.1 | 2.996 | 0.579 | 2.157 | 0.542 | 21.2 | 65.2 | 30.9 | 97.1 | 5.6 | 94.6 | 1.2 | **61.7** |

MA stands for ModelAngelo and DT for DeepTracer. *Calpha RMSD* is the root mean squared deviation of the predicted CA atoms against that of the deposition. *Backbone RMSD* is similar, but for the CA, C, O and N atoms of the protein backbones. *Backbone recall* is the fraction of the deposited residues that were predicted to be within 3 Å (as measured between CA atoms). *Backbone precision* is the fraction of the predicted residues that have a corresponding residue present in the deposition within 3 Å. *Amino acid accuracy* is the fraction of the predicted residues that have a correctly predicted amino acid identity. Finally, completeness is the fraction of deposited residues that were predicted with the correct base annotation. Numbers indicated in boldface are the best in each metric.

**Extended Data Table 2 | Comparison with alternative approaches for the automated building of nucleotides**

| PDB | Resolution (Å) | | Phosphor RMSD (Å) | Backbone RMSD (Å) | Backbone recall | Backbone precision | Base accuracy | Completion |
|---|---|---|---|---|---|---|---|---|
| | | DT | 0.51 | N/A | 86 | 56 | N/A | N/A |
| 7s1g | 2.48 | CR | 1.00 | 1.99 | 68 | 66 | 55 | 37 |
| | | MA | **0.36** | **0.48** | **96** | **99** | **80** | **77** |
| | | DT | 0.86 | N/A | 61 | 38 | N/A | N/A |
| 7zjx | 3.1 | CR | 1.24 | 2.10 | 72 | 60 | 53 | 38 |
| | | MA | **0.48** | **0.61** | **86** | **94** | **66** | **56** |
| | | DT | 0.56 | N/A | 76 | 42 | N/A | N/A |
| 7zpq | 3.47 | CR | 1.14 | 2.05 | 72 | 63 | 52 | 38 |
| | | MA | **0.42** | **0.57** | **92** | **98** | **62** | **57** |

MA stands for ModelAngelo, CR for CryoREAD, and DT for DeepTracer. *Phosphor RMSD* is the root mean squared deviation of the predicted P atoms against that of the deposition. *Backbone RMSD* is similar but for the OP1, P, OP2, and O5' atoms of the nucleotide backbones. *Backbone recall* is the fraction of the deposited residues that were predicted to be within 3 Å (as measured between P atoms). *Backbone precision* is the fraction of predicted residues that have a corresponding residue present in the deposition within 3 Å. *Base accuracy* is the fraction of the predicted residues that have a correctly predicted nucleotide base. Finally, completeness is the fraction of deposited residues that were predicted with the correct base annotation. Numbers indicated in boldface are the best in each metric.

**Extended Data Table 3 | Proteins identified in the *C. reinhardtii* axoneme using ModelAngelo**

| Protein | Phytozome ID | Number of residues | Built residues | EMDB entry | Map resolution (Å) | Location |
|---------|--------------|--------------------|----------------|------------|--------------------|----------|
| RSP24 | Cre08.g800895 | 226 | 1-187 | 22481 | 3.4 | RS2 stalk |
| RSP25 | Cre01.g034550 | 176 | 18-176 | 22475 | 3.2 | RS1 neck* |
| RSP26 | Cre17.g802036 | 128 | 2-125 | 22475 | 3.2 | RS1 neck* |
| RSP27 | Cre05.g240450 | 91 | 36-78 | 22475 | 3.2 | RS1 stalk |
| FAP92 | Cre13.g562250 | 1471 | 1138-1471 | 25381 | 3.8 | C1 microtubule/ Protofilaments 3-4 |
| FAP374 | Cre03.g176600 | 400 | 308-386 | 25381 | 3.8 | C1 microtubule/ Protofilaments 7-9 |

For each identified protein, the phytozome ID is given, together with the number residues in that protein; which residues were built by ModelAngelo; which is the corresponding EMDB entry; the resolution of that map; and the location of the protein. *RSP25 and RSP26 are also expected to occur in the neck of RS2, which is thought to be identical to the neck of RS1.

# Reporting Summary

## Statistics

For all statistical analyses, confirm that the following items are present in the figure legend, table legend, main text, or Methods section.

| n/a | Confirmed | |
|---|---|---|
| ☒ | ☐ | The exact sample size (*n*) for each experimental group/condition, given as a discrete number and unit of measurement |
| ☒ | ☐ | A statement on whether measurements were taken from distinct samples or whether the same sample was measured repeatedly |
| ☒ | ☐ | The statistical test(s) used AND whether they are one- or two-sided<br>*Only common tests should be described solely by name; describe more complex techniques in the Methods section.* |
| ☒ | ☐ | A description of all covariates tested |
| ☒ | ☐ | A description of any assumptions or corrections, such as tests of normality and adjustment for multiple comparisons |
| ☒ | ☐ | A full description of the statistical parameters including central tendency (e.g. means) or other basic estimates (e.g. regression coefficient) AND variation (e.g. standard deviation) or associated estimates of uncertainty (e.g. confidence intervals) |
| ☒ | ☐ | For null hypothesis testing, the test statistic (e.g. *F*, *t*, *r*) with confidence intervals, effect sizes, degrees of freedom and *P* value noted<br>*Give P values as exact values whenever suitable.* |
| ☒ | ☐ | For Bayesian analysis, information on the choice of priors and Markov chain Monte Carlo settings |
| ☒ | ☐ | For hierarchical and complex designs, identification of the appropriate level for tests and full reporting of outcomes |
| ☒ | ☐ | Estimates of effect sizes (e.g. Cohen's *d*, Pearson's *r*), indicating how they were calculated |

*Our web collection on statistics for biologists contains articles on many of the points above.*

## Software and code

Policy information about availability of computer code

| | |
|---|---|
| Data collection | Only publicly available data from the EMDB and PDB were used in this study. |
| Data analysis | ModelAngelo is distributed freely under an MIT licence. We used version 1.0. |

For manuscripts utilizing custom algorithms or software that are central to the research but not yet described in published literature, software must be made available to editors and reviewers. We strongly encourage code deposition in a community repository (e.g. GitHub). See the Nature Portfolio guidelines for submitting code & software for further information.

## Data

Policy information about availability of data

All manuscripts must include a data availability statement. This statement should provide the following information, where applicable:

- Accession codes, unique identifiers, or web links for publicly available datasets
- A description of any restrictions on data availability
- For clinical datasets or third party data, please ensure that the statement adheres to our policy

We only used publicly available data from the EMDB and the PDB.

## Research involving human participants, their data, or biological material

Policy information about studies with human participants or human data. See also policy information about sex, gender (identity/presentation), and sexual orientation and race, ethnicity and racism.

| | |
|---|---|
| Reporting on sex and gender | NA |
| Reporting on race, ethnicity, or other socially relevant groupings | NA |
| Population characteristics | NA |
| Recruitment | NA |
| Ethics oversight | NA |

Note that full information on the approval of the study protocol must also be provided in the manuscript.

# Field-specific reporting

Please select the one below that is the best fit for your research. If you are not sure, read the appropriate sections before making your selection.

☒ Life sciences ☐ Behavioural & social sciences ☐ Ecological, evolutionary & environmental sciences

For a reference copy of the document with all sections, see nature.com/documents/nr-reporting-summary-flat.pdf

# Life sciences study design

All studies must disclose on these points even when the disclosure is negative.

| | |
|---|---|
| Sample size | We used all available structures from the EMDB/PDB, applying exclusion criteria to avoid homologous structures between the training and test sets. |
| Data exclusions | We excluded structures with more than 10% sequence homology between the test and training sets. |
| Replication | NA |
| Randomization | NA |
| Blinding | NA |

# Reporting for specific materials, systems and methods

We require information from authors about some types of materials, experimental systems and methods used in many studies. Here, indicate whether each material, system or method listed is relevant to your study. If you are not sure if a list item applies to your research, read the appropriate section before selecting a response.

## Materials & experimental systems

| n/a | Involved in the study |
|---|---|
| ☒ | ☐ Antibodies |
| ☒ | ☐ Eukaryotic cell lines |
| ☒ | ☐ Palaeontology and archaeology |
| ☒ | ☐ Animals and other organisms |
| ☒ | ☐ Clinical data |
| ☒ | ☐ Dual use research of concern |
| ☒ | ☐ Plants |

## Methods

| n/a | Involved in the study |
|---|---|
| ☒ | ☐ ChIP-seq |
| ☒ | ☐ Flow cytometry |
| ☒ | ☐ MRI-based neuroimaging |

