## [Peer Review File · Nature]

Manuscript Title: Automated model building and protein identification in cryo-EM maps

Reviewer Comments & Author Rebuttals

Reviewer Reports on the Initial Version:

Referees' comments:

Referee #1 (Remarks to the Author):

This manuscript describes ModelAngelo, an extremely impressive piece of work that in my opinion is destined to become the new workhorse tool for automatic building of preliminary atomic models into cryoEM maps. As demonstrated by the authors (and backed up by my own trials) it easily outperforms the previous state of the art in speed, accuracy, and ability to handle enormous datasets of the sort that are rapidly becoming more common. I expect it to have very high impact, both immediately in its application to experimental datasets, and in future as the inspiration and/or foundation for even higher-performing algorithms. Nevertheless, I do have some minor concerns that I believe need to be addressed prior to publication.

1. Page 4, final sentence of introduction: "... and produces near-complete atomic models of comparable quality to those obtained by human experts." Language like this makes me extremely uncomfortable, since it can easily lead the naïve reader into the dangerous belief that automatic building and refinement is sufficient (i.e. that manual checking/correction of a model built by ModelAngelo is generally unnecessary). Figure 1 shows one example from a test run of ModelAngelo 1.0 where this would lead to the retention of a serious error in the model. While it is of course impossible to prove this contention, my strong impression from inspecting a great many recent depositions is that a certain "blind faith" in existing automated tools is already distressingly common. While I appreciate that the need for manual checking is addressed elsewhere in the manuscript, the wording of the introduction has an outsized influence on the take-away message. As such, I would prefer to see this sentence revised to be more nuanced – emphasizing that most models will still contain errors that are unlikely to be resolved through automated means, and that careful manual intervention is still necessary to maximise both correctness and completeness of the model.
2. Tangentially related to the above, I was very surprised to find **no** mention in the text about ModelAngelo's encoding of local confidence into the B-factor column of the model. While I realise this is described in ref. 28, to me this is an absolutely critical piece of information that needs to be restated and discussed in this manuscript.
3. From inspection of the result of my 7uxa test run, it appears to me that the confidence score only encodes confidence in local *geometric* correctness, not *sequence* correctness. For example, for the site in Figure 1 Gly108 has a confidence score of zero (as it should) – but Arg109 has a score of 100 despite being modelled in the position belonging to Gln110. Elsewhere in the model, large stretches of RNA have been given high-perfect confidence scores while being modelled as poly-C (note: the RNA sequence *was* provided as a .fasta file). Indeed the backbone overlay between these stretches and the deposited model is near-perfect. I understand the difficulty in assigning nucleotide bases (and this limitation is well described in the manuscript), and I can see that it might be difficult to combine these qualitatively different concepts of confidence into a single score... might it be sensible for ModelAngelo to supplement its output with some separate per-residue measure of sequence-assignment confidence?

4. Many of the residues assigned zero confidence have exceedingly bad geometry (distorted/stretched bonds as in Figure 1; severely clashing and occasionally interlinked nucleotide bases; etc.). I can understand keeping these in the “raw” output file, but is it really sensible to keep them in the final output.cif?
5. 7γ5e contains numerous large ligands, including about 1500 (!) phycoerythrobilins which are each covalently bonded to one or two cysteine residues (although one of the problems with the deposited model is that these covalent bonds were not modelled). These create branches of strong, somewhat peptide-like density connected directly to the protein – the sort of feature that tends to cause horrible messes in most existing auto-building programs. I am very curious to see how ModelAngelo performs around these sites (but not curious enough to run it on this beast for myself!) – successful avoidance of these pitfalls would in my opinion be one of its strongest selling points.

Figure 1. Left: ModelAngelo model for chain D of 7uxa. Middle: deposited model. ModelAngelo failed to build Gln110 and modelled Arg109 in its place. Naïve refinement of this result (right) leads to a model that appears to fit the density reasonably, albeit with a *cis* peptide bond between Gly108 and Arg109. Notably, ModelAngelo modelled Gly108 with a confidence score of zero – a good illustration of the importance of these scores in assessing the model result.

Referee #2 (Remarks to the Author):

This paper presented ModelAngelo, a machine-learning approach for automated atomic model building in cryo-EM maps. By combining information from the cryo-EM map with information from protein sequence and structure in a single graph neural network, ModelAngelo can build atomic models for proteins, nucleic acids.

The method uses two separate neural networks, the one which predicts C alpha positions in the cryo-EM map, followed by another network that builds up the protein/nucleic acid structures. The architecture of the network is interesting. However, this paper is very similar to the authors' previous publication in The proceedings of the International Conference on Learning Representations (ICLR) 2023. Although this new version can model nucleic acid structure, I do not think the novelty is substantial enough for publication in Nature. Also, the paper lacks objective comparison with other existing methods. Many key quantitative data to clarify and discuss the technical details of modeling ability is lacking.

It seems like the authors are not familiar with rigorous comparison of structure modeling methods. The paper is immature for publication, particularly for Nature.

Major comments

1. This paper used deep learning to detect Ca positions in proteins. This step is not new and has been widely adopted in this area, such as DeepTracer. Compared to before, this paper proposed a GNN to incorporate cryo-EM, sequence information for atomic structure modeling, which is quite new. However, this novelty have been published in ICLR conferences by the authors. The author mainly upgraded the algorithm by simply adding the ability to build nucleotides and an updated Hidden Markov Model algorithm. Compared to the 1st version, there is no big changes and main framework and algorithms are the same as before. Therefore, there is little novel ideas brought by this paper.

2. The paper did not include a full benchmark of model_angelo. Typically, automated structure modeling should measure metrics, such as backbone coverage, sequence recall and RMSD (or TM-score). some of these metrics are included in phenix.chain_comparison (https://phenix-online.org/documentation/reference/chain_comparison.html). I visually checked a few modeled examples in the attachment and compared with the native structures in the database. The modeled structure did not have a very high backbone coverage, particularly for ribosomes. The model completeness is far from satisfactory, not to mention to compare against with human experts. In my impression, the performance seems to be similar to DeepTracer. Moreover, when it comes to multi-chain structure modeling, the chain assignment and sequence consistency is very important to understand biological functions. Such data need to be provided.

3. The ablation studies not provided. For example, the Ca detection accuracy before and after the GNN should be analyzed. Additionally, in the GNN, each module's contribution to the final performances is unclear. The experiments should be carried with/without cryo-EM module (to see if the pre-processed node feature is enough for structure modeling), with/without sequence module, with/without recycling, with/without IPA. I think the overall framework is complicated, thus it is very important to understand what really contributes to the performances. Last but not least, the author mentioned HMM post-

processing for better performances, where I think ablation studies are also needed to understand how much it contributes to the overall performance.

4. The performance needs to be compared with current automated protein structure modeling methods, such as DeepTracer, CR-I-TASSER, DEMO-EM, Buccaneer. For nucleotides structure modeling, DeepTracer, Phenix, and CryoREAD(<https://acs.digitellinc.com/sessions/566247/view>) should be compared to benchmark the nucleotide structure performances.

Comparison needs to be done for single-chain protein targets, hetero and homo multi-chain targets. I would suggest the authors to learn from the CR-I-TASSER paper, for example, to understand how the structure prediction/modeling benchmark should be performed.

5 There is no sequence identity check for nucleotides for building training and testing dataset. That may include some similar nucleotides structures in the testing set which has been seen in training dataset. Then the current nucleotide testing performances may be biased.

6. Also, very importantly, this version of the method should be compared with the previous version that is published in ICLR for protein structure targets. Compare the current/previous versions in terms of Calpha position prediction, RMSD, sequence level accuracy, Calpha coverage etc. We should see here clear improvement by the current method over the previous version.

Actually, we did compare the 2 versions on a dozen protein targets and surprisingly found that the current version's performance seem to be worse than the previous method.

7. In Figures 2a and b, the authors used the Q-score to refer to the local quality of the map. However, the Q-score is defined as the fitness between the deposited model and the map. Although there is a clear correlation between the Q-score and the map resolution, the Q-score also depends on the quality of the deposited model. It should be noted that the value of Q-score is not as commonly used a metric as resolution. The Q-score may make it difficult to understand the quality of the local map. The authors should use properly computed local resolution instead of the Q-score.

8. In the second paragraph on page 10, the authors stated that "Q-scores also correlate with the local resolution".

According to the Q-score paper, the resolution can be estimated from the Q-score, using the equation, [Average Q-score] = -0.178*[resolution] + 1.119 (in Fig5)

<https://www.nature.com/articles/s41592-020-0731-1>

For example, a Q-score of 0.5 corresponds to 3.5 Å resolution.

In Fig2a, model completeness at Q-score of 0.5 is around 50%. Does this imply that ModelAngelo is capable of building only 50% of the deposited model when a map has a 3.5 Å resolution? If it is true, the application of the ModelAngelo is very limited to high-resolution maps. The author needs to address the limitations of ModelAngelo with regards to both local and global resolution.

9. Fig. 2b shows that the most of residues in the deposited models have a Q-score of around 0.7. This

means that most of the maps in the test set have 2.4 Å resolution. I believe that ModelAngelo's performance cannot be accurately demonstrated with a resolution of around 2.4 Å, as it is too high. If the applicable resolution range of the ModelAngelo is around 2.4 Å, please clearly mention it in the manuscript. The information presented in Fig. 2b is misleading because it shows that most of the residues in the deposited models of high-resolution maps have a Q-score of 0.7. Additionally, please provide the EMDb entries and their respective resolutions for the test set.

10. Appropriate use of statistics and treatment of uncertainties

I think the statistical analysis should be carried between model_angelo q-score and deposited q-score, as well as model_angelo FSC and deposited FSC.

11. In the Results section, the authors did not provide information about the performance of amino acid type prediction. Please provide and discuss it.

12. Similarly, what is the accuracy of base prediction for nucleic acid modeling?

13. More data is needed to show the performance of sequence database search. It should also be compared with similar methods, such as FindMySequence.

14. The results of the modeling would critically depend on the accuracy of the initial Calpha atom detection (or nucleotide detection) step, which is done separately from the rest of the modeling process. How much are the final Calpha atom (nucleotide) positions improved in the final model from the initial detection? Please show the data.

To conclude, although the method could be potentially interesting, the work is immature for presentation because it is not compared with similar methods and benchmarked properly (i.e. objectively and quantitatively on a large testing data that are clearly different from training data).

Referee #3 (Remarks to the Author):

Despite recent advances in cryoEM sample preparation, both microscope and camera hardware, and data collection and processing strategies, accurate atomic model building remains a considerable hurdle fraught with human-derived errors; even in the age of AlphaFold predictions. ModelAngelo presents a new machine-learning based automated atomic model building software against cryoEM densities that, as the authors claim, performs as well or better than the author-deposited models. The manuscript is well-written, but I do have some general questions that could potentially improve applicability and adoption.

The authors provide evidence that ModelAngelo provides “accurate” atomic models for most of the available cryoEM maps deposited prior to April 2022. Although these encompass a wide variety of both composition and complexity, I wonder if the authors can specifically elaborate on the following (which also happens to be a limitation of other de novo model builders (e.g. ARPwARP and others):

How does ModelAngelo handle proteins with large cofactors (e.g., ATP or heme) or other non-protein/non-nucleic acids components (e.g. lipids)? For instance, does the heme of hemoglobin promote incorrect backbone tracing? Or how about other larger metal cofactors (e.g., those of the respiratory complexes or P450s)? Does the presence of large cofactors generally cause issues with backbone connectivity? How about integral membrane proteins and tracing of transmembrane helices?

For the examples presented in the manuscript, how well does ModelAngelo handle the 3 thio-linked phycocyanobilin (PCB) cofactor molecules in the phycobilisome? A zoom in of these regions would be more informative than global statistics.

The authors compare the quality (e.g., Q-score) outputs from ModelAngelo to those of the deposited models. Although ModelAngelo performs quite well, I wonder how do the predicted Q scores for each predicted model would compare to what was deposited when each model has been subjected to a single round of all atom refinement (i.e., Phenix refine or similar)? Did the authors consider other geometric quality metrics besides Q-score and Model-Map FSC?

The authors comment on how well ModelAngelo performs on deposited maps but there is a lack of further analyses on why certain maps perform more poorly, besides nominal resolution. I wonder if, for maps that perform more poorly than the average, are there improvements when using a locally filtered map versus those sharpened using a single B-factor? Or how about those processed using programs such as DeepEMHancer? There is little information on map quality correlation to prediction besides nominal resolution (which has its limitation). If there are map modifications that can be performed to improve model accuracy or completeness (or both) then those recommendations are highly welcomed. Perhaps in future iterations the inputs could be the unfiltered half maps and ModelAngelo can handle both sharpening and building.

There are examples in the EMDB of single (or multiple) point mutants that have been determined to high-resolution. If provided the Wt sequence (or no sequence at all), how well does ModelAngelo perform? Does it predict the mutation accurately compared to Wt? If not, at what nominal resolution does this not hold true?

I ran ModelAngelo of a novel, unpublished structure of a heterodimeric protein complex (~120 kDa) at 3.2 Å resolution (nominal) and it correctly identified the domain-swapped organization that other predictors or automated builders failed on (screenshot below). There was one region of lower local resolution where ModelAngelo performed very poorly (in red).

-Mark Herzik

Author Rebuttals to Initial Comments:

ModelAngelo rebuttal

We thank the three referees for their time and their insightful comments, which we feel have led to an improved manuscript. We address their comments in blue below.

Referee #1 (Remarks to the Author):

This manuscript describes ModelAngelo, an extremely impressive piece of work that in my opinion is destined to become the new workhorse tool for automatic building of preliminary atomic models into cryoEM maps. As demonstrated by the authors (and backed up by my own trials) it easily outperforms the previous state of the art in speed, accuracy, and ability to handle enormous datasets of the sort that are rapidly becoming more common. I expect it to have very high impact, both immediately in its application to experimental datasets, and in future as the inspiration and/or foundation for even higher-performing algorithms.

Nevertheless, I do have some minor concerns that I believe need to be addressed prior to publication.

1. Page 4, final sentence of introduction: "... and produces near-complete atomic models of comparable quality to those obtained by human experts." Language like this makes me extremely uncomfortable, since it can easily lead the naïve reader into the dangerous belief that automatic building and refinement is sufficient (i.e. that manual checking/correction of a model built by ModelAngelo is generally unnecessary). Figure 1 (in the attached Word document) shows one example from a test run of ModelAngelo 1.0 where this would lead to the retention of a serious error in the model. While it is of course impossible to prove this contention, my strong impression from inspecting a great many recent depositions is that a certain "blind faith" in existing automated tools is already distressingly common. While I appreciate that the need for manual checking is addressed elsewhere in the manuscript, the wording of the introduction has an outsized influence on the take-away message. As such, I would prefer to see this sentence revised to be more nuanced – emphasizing that most models will still contain errors that are unlikely to be resolved through automated means, and that careful manual intervention is still necessary to maximise both correctness and completeness of the model.

We have modified this sentence as follows:

"Although subsequent error checking and refinement remain necessary, ModelAngelo outperforms human experts in identifying unknown proteins and produces initial atomic models of comparable completeness as those obtained by human experts."

In addition, we now mention in the Discussion:

"Still, some degree of human supervision and intervention will remain necessary. Models from ModelAngelo will still need refinement, for example in Servalcat or Phenix, to optimise their stereochemistry and fit to the cryo-EM map. Users are also strongly encouraged to

manually check the output of ModelAngelo, particularly for those parts of cryo-EM maps with resolutions worse than 3.5-4.0 Å, and rigid body fitting of known domains or connecting loops in lower-resolution map regions to obtain a more complete model falls outside the scope of ModelAngelo. Colouring the model by its predicted confidence in backbone geometry, as stored in the B-factor column of the coordinate file, may guide the user towards parts of the model that are less reliable.”

2. Tangentially related to the above, I was very surprised to find no mention in the text about ModelAngelo’s encoding of local confidence into the B-factor column of the model. While I realise this is described in ref. 28, to me this is an absolutely critical piece of information that needs to be restated and discussed in this manuscript.

Besides the sentence about the B-factor column in the Discussion described above, we have also modified the “Post-processing” subsection, which now reads:

“They [the feature vectors] are also used to predict a confidence score for each residue, which is based on the network’s predicted root-mean-square deviation (RMSD) for the backbone atoms with the deposited structure.”

and

“The predicted backbone RMSD values are mapped to a score between 0 and 1, corresponding to a linear range for RMSDs between 1.2 and 0.5 Å, respectively. This score is stored in the B-factor column of the output coordinate file as a measure of local confidence in the backbone geometry.”

3. From inspection of the result of my 7uxa test run, it appears to me that the confidence score only encodes confidence in local geometric correctness, not sequence correctness. For example, for the site in Figure 1 Gly108 has a confidence score of zero (as it should) – but Arg109 has a score of 100 despite being modelled in the position belonging to Gln110. Elsewhere in the model, large stretches of RNA have been given high-perfect confidence scores while being modelled as poly-C (note: the RNA sequence was provided as a .fasta file). Indeed the backbone overlay between these stretches and the deposited model is near-perfect. I understand the difficulty in assigning nucleotide bases (and this limitation is well described in the manuscript), and I can see that it might be difficult to combine these qualitatively different concepts of confidence into a single score... might it be sensible for ModelAngelo to supplement its output with some separate per-residue measure of sequence-assignment confidence?

The confidence score by ModelAngelo is explicitly meant to predict local geometrical correctness. However, to encode sequence correctness we could also include the entropy in the predicted probabilities of the residues, which would correspond to ModelAngelo’s confidence in its sequence outputs. Although it is not straightforward to combine these, we will explore this for inclusion in a future version of ModelAngelo. We are grateful to the reviewer for raising this point.

4. Many of the residues assigned zero confidence have exceedingly bad geometry (distorted/stretched bonds as in Figure 1; severely clashing and occasionally interlinked

nucleotide bases; etc.). I can understand keeping these in the “raw” output file, but is it really sensible to keep them in the final output.cif?

Pruning is currently based on sequence assignments and minimum chain length, because we assume that the model will always need subsequent refinement, for example with Servalcat. Thereby, regions with poor local geometry may still be useful for the biologist if they are corrected in refinement. Based on the referee’s comment, we will consider the implementation of an additional option to also prune based on local geometric confidence for a future version of ModelAngelo.

5. 7y5e contains numerous large ligands, including about 1500 (!!) phycoerythrobilins which are each covalently bonded to one or two cysteine residues (although one of the problems with the deposited model is that these covalent bonds were not modelled). These create branches of strong, somewhat peptide-like density connected directly to the protein – the sort of feature that tends to cause horrible messes in most existing auto-building programs. I am very curious to see how ModelAngelo performs around these sites (but not curious enough to run it on this beast for myself!) – successful avoidance of these pitfalls would in my opinion be one of its strongest selling points.

ModelAngelo has been trained to build protein and nucleotide residues, but not cofactors. Still, the cryo-EM maps ModelAngelo was trained on did contain cofactor densities. Therefore, ModelAngelo has been incentivized to ignore cofactor densities. In our experience, this is what often happens. To illustrate this, we have included Extended Data Figure 3 for an example of phycocyanobilin and phycoerythrobilin in the phycobilisome structure. The main text now reads:

“ModelAngelo did not attempt to build amino acid or nucleotide residues in the densities for phycocyanobilin or phycoerythrobilin cofactors (Extended Data Figure 3). Because the cryo-EM maps ModelAngelo was trained on did contain cofactor densities, but it was trained to build protein and nucleic acid residues, ModelAngelo has been incentivized to ignore cofactor densities.”

PS: The phycobilisome model built by ModelAngelo is part of the supporting information that we submitted with the original version of the paper, so the referee would not need to build this beast themselves.

Referee #2 (Remarks to the Author):

This paper presented ModelAngelo, a machine- learning approach for automated atomic model building in cryo-EM maps. By combining information from the cryo-EM map with information from protein sequence and structure in a single graph neural network, ModelAngelo can build atomic models for proteins, nucleic acids.

The method uses two separate neural networks, the one which predicts C alpha positions in the cryo-EM map, followed by another network that builds up the protein/nucleic acid structures. The architecture of the network is interesting. However, this paper is very similar to the authors’ previous publication in The proceedings of the International Conference on

Learning Representations (ICLR) 2023. Although this new version can model nucleic acid structure, I do not think the novelty is substantial enough for publication in Nature.

Also, the paper lacks objective comparison with other existing methods. Many key quantitative data to clarify and discuss the technical details of modeling ability is lacking.

It seems like the authors are not familiar with rigorous comparison of structure modeling methods. The paper is immature for publication, particularly for Nature.

Major comments

1. This paper used deep learning to detect Ca positions in proteins. This step is not new and has been widely adopted in this area, such as DeepTracer. Compared to before, this paper proposed a GNN to incorporate cryo-EM, sequence information for atomic structure modeling, which is quite new. However, this novelty have been published in ICLR conferences by the authors. The author mainly upgraded the algorithm by simply adding the ability to build nucleotides and an updated Hidden Markov Model algorithm. Compared to the 1st version, there is no big changes and main framework and algorithms are the same as before. Therefore, there is little novel ideas brought by this paper.

As confirmed by the editor, publication in conference proceedings is allowed by Nature, provided there exists “a substantial extension of results, methodology, analysis, conclusions and/or implications”. Our ICLR proceedings introduced an unfinished beta version of ModelAngelo, without functionality for the identification of unknown proteins, and with a much less extensive analysis of its model building capabilities.

2. The paper did not include a full benchmark of model_angelo. Typically, automated structure modeling should measure metrics, such as backbone coverage, sequence recall and RMSD (or TM-score). some of these metrics are included in phenix.chain_comparison (https://phenix-online.org/documentation/reference/chain_comparison.html). I visually checked a few modeled examples in the attachment and compared with the native structures in the database. The modeled structure did not have a very high backbone coverage, particularly for ribosomes. The model completeness is far from satisfactory, not to mention to compare against with human experts. In my impression, the performance seems to be similar to DeepTracer. Moreover, when it comes to multi-chain structure modeling, the chain assignment and sequence consistency is very important to understand biological functions. Such data need to be provided.

Our original manuscript assessed ModelAngelo on all structures that were deposited in the EMDB over an 11 months period, with less than 30,000 residues and less than 10% sequence identity to any protein in the training set, excluding icosahedral viruses. We are not aware that such an elaborate benchmark has ever been used to test other model building programs. In Figure 2a, the completeness (shown in blue) informs on both “backbone coverage” and “sequence recall” for this data set, as only correctly placed residues with the correct sequence assignment are included. Backbone RMSD is shown in pink.

To address the referee’s comments 2-4 and 6-9, we have expanded our initial benchmark with a smaller subset of 27 protein structures. Following the referee’s suggestion, 9 structures are single-chain entries, 9 structures are homo-oligomers and 9 structures are

hetero-oligomers. For each group of 9 structures, 3 had overall resolutions below 3.3 Å; 3 ranged from 3.3-2.8 Å; and 3 had resolutions better than 2.8 Å. Because we only chose structures for which unfiltered half-maps were available for download from the EMDB, we could use ResMap to calculate local resolution estimates for all deposited residues and measure the performance of ModelAngelo and four alternative programs (DeepTracer, Buccaneer, Phenix and Demo-EM) at varying local resolutions in the maps. We also ran phenix.chain_comparison to analyse the results. These tests demonstrate that ModelAngelo outperforms the alternative approaches. The new results are shown in Figure 2e and Extended Data Table 1, and they are described in a new section:

“ModelAngelo outperforms alternative approaches *In a second test, we compared the performance of ModelAngelo with existing approaches for automated model building in cryo-EM maps. For this test, we used a subset of 27 protein structures from the 177 structures described above. We selected nine single-chain structures, nine homo-oligomeric structures, and nine hetero-oligomeric structures. For each of these types of structures, we selected three structures with overall resolutions below 3.3 Å, three structures between 3.3 and 2.8 Å, and three structures with resolutions better than 2.8 Å. For all 27 structures, unfiltered half-maps were available for download from the EMDB, and we used these to calculate local resolutions in ResMap (40). We then used Phenix (41), Demo-EM (42), Buccaneer (20), and DeepTracer (21) for automated model building in these maps and compared the completeness of the resulting models with those obtained using ModelAngelo (Figure 2e and Extended Data Table 1). The best alternative approach, DeepTracer, built approximately 80% of the deposited residues in regions of the maps with local resolutions in the range of 2.5-3 Å; the remaining approaches built models with considerably lower completeness. In contrast, ModelAngelo built up to 80% of the deposited residues in regions of the maps with local resolutions down to 3.5-4 Å, reflecting the observation that manual building by human experts also becomes prone to errors at resolutions below 4 Å.”*

3. The ablation studies not provided. For example, the Ca detection accuracy before and after the GNN should be analyzed. Additionally, in the GNN, each module's contribution to the final performances is unclear. The experiments should be carried with/without cryo-EM module (to see if the pre-processed node feature is enough for structure modeling), with/without sequence module, with/without recycling, with/without IPA. I think the overall framework is complicated, thus it is very important to understand what really contributes to the performances. Last but not least, the author mentioned HMM post-processing for better performances, where I think ablation studies are also needed to understand how much it contributes to the overall performance.

We performed the requested ablation studies on the new test set described under comment 2, but note that it is not possible to run ModelAngelo without its cryo-EM module. The results of the new ablation study are shown in Figure 2f. They confirm our previous observations from the ICLR paper that all three modules are necessary. The section mentioned above ends with:

“Tests where we ran ModelAngelo without one or more of its modules, indicate that its performance comes from a combination of all three modules (Figure 2f), which is in accordance with previous observations (28).”

4. The performance needs to be compared with current automated protein structure modeling methods, such as DeepTracer, CR-I-TASSER, DEMO-EM, Buccaneer. For nucleotides structure modeling, DeepTracer, Phenix, and CryoREAD (<https://acs.digitellinc.com/sessions/566247/view>) should be compared to benchmark the nucleotide structure performances.

Comparison needs to be done for single-chain protein targets, hetero and homo multi-chain targets. I would suggest the authors to learn from the CR-I-TASSER paper, for example, to understand how the structure prediction/modeling benchmark should be performed.

See our reply to point 2 above. Using the new test set, we have included a comparison of ModelAngelo with DeepTracer, Demo-EM, Buccaneer and Phenix. For three of the ribosome structures, we also compared our results for building nucleotides with those from cryoREAD. We also tried to run CR-I-TASSER, but their server did not return any solutions, and attempts to install the program locally failed repeatedly.

5 There is no sequence identity check for nucleotides for building training and testing dataset. That may include some similar nucleotides structures in the testing set which has been seen in training dataset. Then the current nucleotide testing performances may be biased.

There are much fewer nucleotide structures in the EMDB than there are protein structures. Therefore, in order to allow training of deep neural networks, it is not possible to remove all non-unique nucleotide sequences like we did for proteins. This problem will also exist for other automated model building programs.

6. Also, very importantly, this version of the method should be compared with the previous version that is published in ICLR for protein structure targets. Compare the current/previous versions in terms of Calpha position prediction, RMSD, sequence level accuracy, Calpha coverage etc. We should see here clear improvement by the current method over the previous version. Actually, we did compare the 2 versions on a dozen protein targets and surprisingly found that the current version's performance seem to be worse than the previous method.

Although our experience does not reflect that of the referee, we cannot guarantee that ModelAngelo 1.0 will always perform better than the beta version. However, the latter will not be supported in the future. The current manuscript describes the implementation that people will be using and citing.

7. In Figures 2a and b, the authors used the Q-score to refer to the local quality of the map. However, the Q-score is defined as the fitness between the deposited model and the map. Although there is a clear correlation between the Q-score and the map resolution, the Q-score also depends on the quality of the deposited model. It should be noted that the value of Q-score is not as commonly used a metric as resolution. The Q-score may make it difficult to understand the quality of the local map. The authors should use properly computed local resolution instead of the Q-score.

8. In the second paragraph on page 10, the authors stated that "Q-scores also correlate with the local resolution".

According to the Q-score paper, the resolution can be estimated from the Q-score, using the equation, [Average Q-score] = -0.178*[resolution] + 1.119 (in Fig5)]

For example, a Q-score of 0.5 corresponds to 3.5 Å resolution. In Fig2a, model completeness at Q-score of 0.5 is around 50%. Does this imply that ModelAngelo is capable of building only 50% of the deposited model when a map has a 3.5 Å resolution? If it is true, the application of the ModelAngelo is very limited to high-resolution maps. The author needs to address the limitations of ModelAngelo with regards to both local and global resolution.

In response to the referee's comments, we chose the 27 structures for our new test set also based on the availability of their half-maps, which we used to calculate local resolution estimates in ResMap. (We use Q-scores as a proxy for local map quality for the larger test set because half-maps are not available for all of those structures). For the 27 structures in the new test, we now report the completeness of ModelAngelo and alternative model building programs for six local resolution bins (Figure 2e). ModelAngelo builds approximately 80% of all residues in the local resolution bin of 3.5-4 Å.

Below we show the distribution of Q-scores (in %) for the 27 structures within each of the 6 local-resolution bins that we used in our analyses. This plot confirms the overall trend of Q-score versus local resolution, which was already pointed out in the original Q-score paper by Pintillie et al.

9. Fig. 2b shows that the most of residues in the deposited models have a Q-score of around 0.7. This means that most of the maps in the test set have 2.4 Å resolution. I believe that

ModelAngelo's performance cannot be accurately demonstrated with a resolution of around 2.4 Å, as it is too high. If the applicable resolution range of the ModelAngelo is around 2.4 Å, please clearly mention it in the manuscript. The information presented in Fig. 2b is misleading because it shows that most of the residues in the deposited models of high-resolution maps have a Q-score of 0.7. Additionally, please provide the EMDB entries and their respective resolutions for the test set.

We do not understand what the referee means by "it is too high". The original test set was derived for all of the EMDB submissions from Apr-2022 until Feb 2023 (removing entries with more than 30,000 residues, icosahedral viruses and entries with more than 10% sequence identity in the training set). As such, our set of structures is representative of last year's deposited cryo-EM structures. Figure 2b therefore depicts what is relevant to the modern experimentalist.

The supplement now also contains the EMDB entries and the overall reported resolutions, which range from 1.5 to 4.0 Å, with an average of 3.1 Å.

10. Appropriate use of statistics and treatment of uncertainties

I think the statistical analysis should be carried between model_angelo q-score and deposited q-score, as well as model_angelo FSC and deposited FSC.

Based on the referee's comment, we performed statistical analysis on the distributions of per-residue Q-scores of the models from ModelAngelo versus those from the deposited models. The R² for linear regression was 0.55. Because these distributions are not normally distributed, we ran a Mann-Whitney U test to see if the distributions are different. The p-value is 1.36 e⁻³⁹, indicating that the two distributions, which comprise 300,000 points, are statistically different. For the FSCs, the R² of linear regression is 0.982. Again, the statistical test (a paired t-test) had a p-value of 0.0046, suggesting that those two distributions are also different. However, the table below illustrates that the differences are in the second or third decimal of the Q-scores and FSCs, which are probably not relevant to the experimentalist.

Statistic	ModelAngelo Qscore	Deposited Qscore	ModelAngelo FSC	Deposited FSC
mean	0.687159	0.690000	0.514558	0.519458
std	0.126182	0.130599	0.165423	0.166809
min	-0.268090	-0.665264	0.089400	0.091400
25%	0.630259	0.636275	0.393775	0.396600
50%	0.706112	0.709583	0.543550	0.547050
75%	0.763865	0.765594	0.639975	0.642250

max	0.971461	0.971711	0.818400	0.824700
-----	----------	----------	----------	----------

Still, to avoid any misconception in terms of implied statistical relevance, we have modified the sentence that originally read: “A comparison of Q-scores calculated for the models built by ModelAngelo with those calculated for the deposited models shows that models from ModelAngelo are as good as the deposited ones (Figure 2c)”, by replacing the words “as good as” with “of similar quality to”.

11. In the Results section, the authors did not provide information about the performance of amino acid type prediction. Please provide and discuss it.

12. Similarly, what is the accuracy of base prediction for nucleic acid modeling?

Model building in ModelAngelo does not rely on discrete amino acid or nucleotide predictions of individual residues. Instead, it feeds probability vectors for all twenty amino acids or eight types of nucleotides into HMM searches to assign entire chains. Whereas individual residue prediction may represent a key step in alternative programs, in ModelAngelo the key output is the identified chain. Our completeness metric measures the accuracy of the chain assignments.

13. More data is needed to show the performance of sequence database search. It should also be compared with similar methods, such as FindMySequence.

In the revised manuscript, we now report the results of FindMySequence for both the phycobilisome and the radial spoke/central apparatus structures. In total, FindMySequence identified 2 of the 12 unknown proteins that are described in the manuscript. The text for the phycobilisome now reads:

“Using the backbone traces in the deposited model, findMySequence identified only two of the unassigned proteins (Psb34 and PsbW).”

And for the radial spoke and CA structures:

“Using ModelAngelo's backbone traces, findMySequence was unable to identify any of these proteins.”

14. The results of the modeling would critically depend on the accuracy of the initial Calpha atom detection (or nucleotide detection) step, which is done separately from the rest of the modeling process. How much are the final Calpha atom (nucleotide) positions improved in the final model from the initial detection? Please show the data.

For the 27 structures in the test set, the average RMSD between the coordinates from the Calpha detection and the final prediction is 0.8 Å. This leads to an improvement of the RMSD with the deposited Calpha positions from 0.9 to 0.3 Å. For those interested in the accuracy of this intermediate step, ModelAngelo also outputs the detected Calpha atoms.

To conclude, although the method could be potentially interesting, the work is immature for presentation because it is not compared with similar methods and benchmarked properly (i.e. objectively and quantitatively on a large testing data that are clearly different from training data).

Referee #3 (Remarks to the Author):

Despite recent advances in cryoEM sample preparation, both microscope and camera hardware, and data collection and processing strategies, accurate atomic model building remains a considerable hurdle fraught with human-derived errors; even in the age of AlphaFold predictions. ModelAngelo presents a new machine-learning based automated atomic model building software against cryoEM densities that, as the authors claim, performs as well or better than the author-deposited models. The manuscript is well-written, but I do have some general questions that could potentially improve applicability and adoption.

The authors provide evidence that ModelAngelo provides “accurate” atomic models for most of the available cryoEM maps deposited prior to April 2022. Although these encompass a wide variety of both composition and complexity, I wonder if the authors can specifically elaborate on the following (which also happens to be a limitation of other de novo model builders (e.g. ARPwARP and others):

How does ModelAngelo handle proteins with large cofactors (e.g., ATP or heme) or other non-protein/non-nucleic acids components (e.g. lipids)? For instance, does the heme of hemoglobin promote incorrect backbone tracing? Or how about other larger metal cofactors (e.g., those of the respiratory complexes or P450s)? Does the presence of large cofactors generally cause issues with backbone connectivity? How about integral membrane proteins and tracing of transmembrane helices?

For the examples presented in the manuscript, how well does ModelAngelo handle the 3 thio-linked phycocyanobilin (PCB) cofactor molecules in the phycobilisome? A zoom in of these regions would be more informative than global statistics.

A similar point about cofactor densities was raised by referee #1, please see our reply there.

The authors compare the quality (e.g., Q-score) outputs from ModelAngelo to those of the deposited models. Although ModelAngelo performs quite well, I wonder how do the predicted Q scores for each predicted model would compare to what was deposited when each model has been subjected to a single round of all atom refinement (i.e., Phenix refine or similar)? Did the authors consider other geometric quality metrics besides Q-score and Model-Map FSC?

ModelAngelo is primarily a model building programme and its output still needs to be subjected to refinement with stereochemical restraints. We now mention in the Discussion:

“Models from ModelAngelo will still need refinement, for example in Servalcat or Phenix, to optimise their stereochemistry and fit to the cryo-EM map.”

In the current work, we ran all models from ModelAngelo through a standard refinement cycle in Servalcat. The main text reads:

“The output coordinates from ModelAngelo were refined against the cryo-EM map using a standard refinement cycle in Servalcat, and the refined models were compared to the deposited ones.”

For the 27 protein structures in the new test set, we now report quality metrics as reported by phenix.chain_comparison in Extended Data Table 1.

The authors comment on how well ModelAngelo performs on deposited maps but there is a lack of further analyses on why certain maps perform more poorly, besides nominal resolution. I wonder if, for maps that perform more poorly than the average, are there improvements when using a locally filtered map versus those sharpened using a single B-factor? Or how about those processed using programs such as DeepEMHancer? There is little information on map quality correlation to prediction besides nominal resolution (which has its limitation). If there are map modifications that can be performed to improve model accuracy or completeness (or both) then those recommendations are highly welcomed. Perhaps in future iterations the inputs could be the unfiltered half maps and ModelAngelo can handle both sharpening and building.

ModelAngelo was trained with augmentation through a variety of positive and negative B-factors. It should therefore be relatively stable to local variations in B-factor. The new figure 2e now reports performance of ModelAngelo at different local resolutions for our new test set of 27 protein structures.

It is true that modification of the cryo-EM maps by programs like DeepEMHancer and EMReady could potentially improve the results. However, it would probably be best if ModelAngelo were re-trained, providing such maps as input to ModelAngelo. We have added the following sentence to the Discussion:

“ModelAngelo was trained with augmentation through a variety of positive and negative B-factors. It should therefore be relatively stable to local variations in B-factor. It is possible that combining ModelAngelo with neural networks that make cryo-EM maps look more like proteins, e.g. Sanchez-Garcia et al (2021) and He et al (2023), could lead to further improvements, although this would probably require re-training of ModelAngelo to reach its full potential.”

There are examples in the EMDB of single (or multiple) point mutants that have been determined to high-resolution. If provided the Wt sequence (or no sequence at all), how well does ModelAngelo perform? Does it predict the mutation accurately compared to Wt? If not, at what nominal resolution does this not hold true?

The current implementation is not able to detect and build point mutations in an automated manner. Unexpected point mutations will therefore represent a penalty in the HMM searches to dock chains and may lead to suboptimal models. The user would then need to go in and check why this happened. Manual checking of the output of ModelAngelo is recommended anyway, as now made explicit in the revised manuscript in response to referee #1.

I ran ModelAngelo of a novel, unpublished structure of a heterodimeric protein complex (~120 kDa) at 3.2 Å resolution (nominal) and it correctly identified the domain-swapped organization that other predictors or automated builders failed on (screenshot below). There was one region of lower local resolution where ModelAngelo performed very poorly (in red).

It is great to see ModelAngelo correctly spotted the domain swap. In low-resolution regions of the map, ModelAngelo will make mistakes, as we now explicitly show in Figure 2e. This example provides support for the implementation of a pruning option based on local geometry of the backbone, as discussed for point 4 of referee #1, which is something we will consider.

Reviewer Reports on the First Revision:

Referees' comments:

Referee #1 (Remarks to the Author):

I am satisfied with the changes made to the manuscript in response to my suggestions, and am happy to recommend the revised manuscript for publication.

Referee #2 (Remarks to the Author):

In the revised manuscript, I see some of my comments are addressed but the authors failed or omitted to respond to major questions.

The major issues that are unresolved include rigorous comparison with state-of-the-art methods to show superiority of the current methods over the other methods, convincing ablation study, and lack of novelty of the method in comparison with their earlier publication in a top CS conference paper.

Protein and nucleic acid structure modeling is a very competitive field. Publication of structure modeling paper in Nature would need to show outstanding, convincing performance and novelty, which clearly lacks in the current work. I conducted some of the benchmark studies that were requested but not performed by the authors. However, these studies did not demonstrate strong performance. The weaknesses in the current work are not related to presentation and cannot be rectified. Therefore, I do not recommend considering this work further for Nature.

1. I respectfully disagree with the novelty of this work in consideration with the earlier ICLR proceedings. As responded by the authors in the response letter, only the new functionality added in this work is HMM-based identification of unknown proteins (which is also not a new idea considering other related works). This work included more modeling results than the ICLR proceedings version, but the added results do not add new scientific findings as it is benchmark studies. It has two examples that compares with “human” but only two examples. I do not believe that these additions are sufficient for an additional publication in Nature (I may be more positive if this is for publication in a more specific journal). Most importantly, I observed clear performance drop in protein structure modeling compared to their published version (figure comparison included later).
2. The response by the authors for this comment is unsatisfactory. As I commented in the previous review, “the chain assignment and sequence consistency is very important to understand biological functions. Such data need to be provided.” Since

I noticed that ModelAngelo generated a lot of fragments with different chain ID assignments for some targets, the metric of completeness is not enough to discuss the accuracies of predicted models. To evaluate the connection of C-alpha model and the sequence order, TM-score for each chain is a useful metric. I asked to compute TM-score on the test dataset, which the authors have totally ignored. From other paper's comparison (SmartFold, DeepMainMast), the TM-score of model-angelo is substantially smaller compared to other methods on multiple datasets.

3. For the requested ablation studies, the authors only answered part of the questions while ignoring many important concerns. First, the most important ablation study is missing: the Ca detection accuracy before and after the GNN is missing. Second, with/without cryo-EM module is feasible since you have the initial Ca positions from Unet detection and you have sequence input. Therefore, it is possible to remove cryo-EM modules in GNN. Third, with/without recycling experiment is missing, that is necessary as it is shown in important factor in AlphaFold, and other structure modeling algorithms. Fourth, before/after HMM post-processing is not analyzed.

Since the authors refused follow the requests, I did one of them:

This is comparing the Calpha position accuracy (F1 score) before and after applying GNN on the dataset used in the ICLR paper using the current ModelAngelo (ver. 1.0.0). As shown, the average F1 score decreased from 83.56 to 77.24, and the Calpha detection went worse by adding GNN, which is supposed to be the most important novelty of ModelAngelo compared to existing works.

4. I was totally disappointed that the authors did not compare ModelAngelo with CR-I-TASSER, which is currently considered the best modeling method for cryo-electron microscopy maps. The CR-I-TASSER group provides a test dataset, including experimental EM maps with 2-5Å resolution, at <https://zhanggroup.org/CR-I-TASSER/>. In addition, TM-scores of CR-I-TASSER models are also made available by the authors at <https://www.nature.com/articles/s41592-021-01389-9#Sec22>.

Since the authors did not perform the comparison against CR-I-TASSER, I ran the current version of ModelAngelo on the CR-I-TASSER dataset and compared:

ModelAngelo performed substantially worse than CR-I-TASSER, as shown. ModelAngelo showed lower TM-Score for many cases than CR-I-TASSER with an

average TM-Score of 0.59, while CR-I-TASSER had an average of 0.81.

I also recently found that the DeepMainmast paper currently on bioRxiv also showed a comparison with ModelAngelo, which shows similar results (<https://www.biorxiv.org/content/10.1101/2023.10.19.563151v2>) on three different datasets, including the dataset used in the ModelAngelo paper.

Another recent work, SmartFold (<https://www.biorxiv.org/content/10.1101/2023.11.02.565403v1.full.pdf>), also compares with ModelAngelo, reporting that the ModelAngelo's performance is worse than SmartFold. The SmartFold paper also compared with EMBuild, a recent publication in Nature Communications (<https://www.nature.com/articles/s41467-022-31748-9>), showing EMBuild's better performance than ModelAngelo in the single-chain dataset.

Benchmark values (mean and median) of test samples (N=126)

Method	SeqMatch	ChainMatch	TM-score
SMARTFold	0.858 / 0.933	0.857 / 0.933	0.936 / 0.977
AlphaFold-Multimer	0.686 / 0.773	0.680 / 0.766	0.862 / 0.941
EMBuild	0.852 / 0.914	0.850 / 0.910	0.918 / 0.976
ModelAngelo	0.768 / 0.867	0.751 / 0.848	-
Phenix	0.252 / 0.193	0.219 / 0.157	-

The authors compared DeepTracer, Demo-EM, Buccaneer and Phenix, which I appreciate. But except for Demo-EM, the other three methods are now baseline methods, which any

new method should clearly win, thus not sufficient to claim the state-of-the-art-ness of the method. By the way, for comparing with methods, please make scatter plots, so that the values of individual maps are visible, instead of bar graphs.

Comparison against CryoREAD on only 3 cherry-picked maps and making conclusion out of it is totally inappropriate. I ran CryoREAD and ModelAngelo on the testset of ModelAngelo and CryoREAD datasets. The plot shown below is the sequence recall, which apparently shows substantially better performance by CryoREAD. In terms of average backbone recall, ModelAngelo had 0.842, while CryoREAD had 0.855, slightly higher than ModelAngelo. In terms of sequence recall, the gap is larger, with ModelAngelo had 0.419, while CryoREAD had 0.512. The comparison figure is attached here.

5. Page 10 and Figure 2f, “Tests, where we ran ModelAngelo without one or more of its modules, indicate that its performance comes from a combination of all three modules (Figure 2f), which is in accordance with previous observations [28].” According to Figure 2f, the order of performance of completeness (Sequence Recall) from highest to lowest is ModelAngelo > Removed IPA > Removed Sequence, but figure3C in [28] it is Original > Seq Removed > IPA Removed. Please correct the paragraph on page 10. Based on the data provided, can one assume that the Seq and IPA modules made equal contributions?

6. I mentioned that “*the authors did not provide information about the performance of amino acid type prediction. Please provide and discuss it.*” However, the author declined to reveal the results. I believe that detecting the type of amino acid accurately is informative, especially when the target sequence is unknown. Since the availability of the "model_angelo build_no_seq" command in ModelAngelo, the authors should display the accuracy of amino acid type predictions by considering the residue type with the highest probability.

7. “Similarly, what is the accuracy of base prediction for nucleic acid modeling?” This request was also neglected.

8. It is difficult to get a clear understanding of the overall performance of ModelAngelo for nucleotide targets based on the information presented in Figure 3. It appears that the figure only shows the successful results from the nucleotide targets. Similar to the information provided in Extended Table 1, results for the 11 ribosome structures and CRISPR-associated transpososome targets should be explained in more detail.

9. The authors totally neglected to provide data for this comment from the previous review:

“6. Also, very importantly, this version of the method should be compared with the previous version that is published in ICLR for protein structure targets. Compare the current/previous versions in terms of Calpha position prediction, RMSD, sequence level accuracy, Calpha coverage etc. We should see here clear improvement by the current method over the previous version.

Actually, we did compare the 2 versions on a dozen protein targets and surprisingly found that the current version’s performance seem to be worse than the previous method.”

The author verbally replied that “Although our experience does not reflect that of the referee, we cannot guarantee that ModelAngelo 1.0 will always perform better than the beta version.”

But in our analysis, the current version 1.0 is consistently worse, which questions the justification of publishing this new version in a paper:

10. To this question: "There is no sequence identity check for nucleotides for building training and testing dataset. That may include some similar nucleotides structures in the testing set which has been seen in training dataset. Then the current nucleotide testing performances may be biased." , the authors answered:

"There are much fewer nucleotide structures in the EMDB than there are protein structures. Therefore, in order to allow training of deep neural networks, it is not possible to remove all non-unique nucleotide sequences like we did for proteins. This problem will also exist for other automated model building programs."

This is not an appropriate answer. Regardless of how other people are doing, there are ways to still address this question by showing some data. I know DNA/RNA structure modeling methods that consider sequence redundancy as much as they can.

Overall, I reiterate my clear opinion that I cannot endorse further consideration of this work for publication in Nature. The authors are well respected for their work on the cryo-EM program package, Relion, which I also personally use and appreciate. I would agree that having ModelAngelo in the Relion package may be useful for users. However, the performance of ModelAngelo is not state-of-the-art, with so many incomplete and inappropriate responses in this revision. The current work does not justify publication in Nature.

Referee #3 (Remarks to the Author):

No further suggestions/requests.

Author Rebuttals to First Revision:

We respond to the second round of comments by reviewer #2 in blue below.

In the revised manuscript, I see some of my comments are addressed but the authors failed or omitted to respond to major questions.

The major issues that are unresolved include rigorous comparison with state-of-the-art methods to show superiority of the current methods over the other methods, convincing ablation study, and lack of novelty of the method in comparison with their earlier publication in a top CS conference paper.

Protein and nucleic acid structure modeling is a very competitive field. Publication of structure modeling paper in Nature would need to show outstanding, convincing performance and novelty, which clearly lacks in the current work. I conducted some of the benchmark studies that were requested but not performed by the authors. However, these studies did not demonstrate strong performance. The weaknesses in the current work are not related to presentation and cannot be rectified. Therefore, I do not recommend considering this work further for Nature.

Below, we will show that the “benchmark studies” performed by the reviewer are flawed.

1. I respectfully disagree with the novelty of this work in consideration with the earlier ICLR proceedings. As responded by the authors in the response letter, only the new functionality added in this work is HMM-based identification of unknown proteins (which is also not a new idea considering other related works). This work included more modeling results than the ICLR proceedings version, but the added results do not add new scientific findings as it is benchmark studies. It has two examples that compares with “human” but only two examples. I do not believe that these additions are sufficient for an additional publication in Nature (I may be more positive if this is for publication in a more specific journal). Most importantly, I observed clear performance drop in protein structure modeling compared to their published version (figure comparison included later).

This point was settled by the editor after the first round of review. Moreover, the reviewer ignores the novelty of the identification of unknown proteins in cryo-EM maps.

2. The response by the authors for this comment is unsatisfactory. As I commented in the previous review, “the chain assignment and sequence consistency is very important to understand biological functions. Such data need to be provided.” Since I noticed that ModelAngelo generated a lot of fragments with different chain ID assignments for some targets, the metric of completeness is not enough to discuss the accuracies of predicted models. To evaluate the connection of C-alpha model and the sequence order, TM-score for each chain is a useful metric. I asked to compute TM-score on the test dataset, which the authors have totally ignored. From other paper’s comparison (SmartFold, DeepMainMast), the TM-score of model angelo is substantially smaller compared to other methods on multiple datasets.

The reviewer wrote in their original comments: “Typically, automated structure modeling should measure metrics, such as backbone coverage, sequence recall and RMSD (or TM-score).” In response, we included in Extended Data Tables 1 and 2 of our revision detailed information about backbone coverage, sequence recall (which is included in our ‘completeness’ measure) and backbone RMSD of proteins and nucleotides targets in our test sets. Now, the reviewer focuses on the TM-score, which was only mentioned in parentheses in the original review. As we will show below, TM-score analysis is not straightforward, which is why we chose not to use it in our original rebuttal. Nevertheless, we now show that when performed more carefully, TM-score analysis presents a much more favourable view of ModelAngelo’s performance than the reviewer presents.

3. For the requested ablation studies, the authors only answered part of the questions while ignoring many important concerns. First, the most important ablation study is missing: the Ca detection accuracy before and after the GNN is missing. Second, with/without cryo-EM module is feasible since you have the initial Ca positions from Unet detection and you have sequence input. Therefore, it is possible to remove cryoEM modules in GNN. Third, with/without recycling experiment is missing, that is necessary as it is shown in important factor in AlphaFold, and other structure modeling algorithms. Fourth, before/after HMM post-processing is not analyzed.

ModelAngelo is a program to build atomic models in cryo-EM maps. One of its main strengths is the incorporation of

additional sources of information in a GNN: ESM embeddings through its sequence module, and information about the local geometry of the graph nodes through its IPA module. The usefulness of both additional sources of information was tested in our ablation studies. We think it is less useful to ablate the cryo-EM module, as it removes the main source of input data that the original task is conditioned on: ModelAngelo is not intended as a structure prediction program.

We could add statistics about our models before and after the HMM post-processing to the Extended Data Tables 1 and 2, if the editor (or the other reviewers) deem this to be a useful addition.

Since the authors refused follow the requests, I did one of them:

This is comparing the Calpha position accuracy (F1 score) before and after applying GNN on the dataset used in the ICLR paper using the current ModelAngelo (ver. 1.0.0). As shown, the average F1 score decreased from 83.56 to 77.24, and the Calpha detection went worse by adding GNN, which is supposed to be the most important novelty of ModelAngelo compared to existing works.

We did not refuse to follow this request. In point 14 of our first rebuttal, we reported that over the test set of 27 proteins, ModelAngelo's GNN improves the Calpha RMSD from 0.9 to 0.3 Angstroms. Again, the reviewer's observations do not match ours: the GNN generally improves the positions. Maybe the reviewer is confusing the pruning step of ModelAngelo that occurs after the GNN, where we remove residues that we cannot correspond to the sequence from the output. This will necessarily reduce recall, but improves precision. We have found that users find the removal of less confident parts of the model more useful.

4. I was totally disappointed that the authors did not compare ModelAngelo with CR-I TASSER, which is currently considered the best modeling method for cryo-electron microscopy maps. The CR-I-TASSER group provides a test dataset, including experimental EM maps with 2-5Å resolution, at [https://zhanggroup.org/CR-I TASSER/](https://zhanggroup.org/CR-I-TASSER/). In addition, TM-scores of CR-I-TASSER models are also made available by the authors at <https://www.nature.com/articles/s41592-021-01389-9#Sec22>.

The reason why we did not use the CR-I-TASSER test set is because it is not a straightforward test for model building in cryo-EM maps, as it consists of segmented maps for single-chain proteins. These segmented maps have been carved out of the full cryo-EM maps of larger, multi-chain complexes. Not only does the segmenting of individual chains lead to an overly simplistic problem (the experimentalist needs to build many chains at once, and deciding where each chain is forms a major part of that problem), the neural networks of ModelAngelo were also not trained on segmented maps, which as we show below leads to suboptimal results. But the opposite is true for CR-I-TASSER. Its paper's supplementary information states: "*Although CR-I-TASSER is designed for single-chain protein modeling, we tested it on the experimental dataset with full density maps instead of segmented maps. [...] As shown in Supplementary Figure 7a, the performance of CR-I-TASSER dropped in the full-map modeling with average TMscore=0.670, compared to the TM-score=0.752 with segmented maps.*" This is a substantial drop in performance for the more realistic case of the problem. As a side note, we also point out that training of CR-I-TASSER was stopped based on its performance of the test set, which breaks the basic assumption of an independent test set.

Perhaps the other reviewers can comment too, but in our opinion the impact of CR-I-TASSER on the cryoEM field has been limited. Its paper came out on 2 Feb 2022, and has since been cited 24 times, almost all of which are in the context of machine-learning applications, rather than cryoEM structure determination projects.

Since the authors did not perform the comparison against CR-I-TASSER, I ran the current version of ModelAngelo on the CR-I-TASSER dataset and compared: ModelAngelo performed substantially worse than CR-I-TASSER, as shown. ModelAngelo showed lower TM-Score for many cases than CR-I-TASSER with an average TM-Score of 0.59, while CR-I-TASSER had an average of 0.81.

The design of this comparison with CR-I-TASSER is flawed. Besides the above-mentioned problems with segmented maps of individual chains in the CR-I-TASSER test set, calculating TM-scores for multi-chain complexes is also not straightforward. The TM-score program from the Zhang lab appears to grab the first chain in the input file, but this is not necessarily the best matching one (as the order in the input file is undefined). Instead, we made a modification to this program that includes finding the best matching chain in the ModelAngelo output file by sequence similarity.

Also, we do not agree that TM-scores are more indicative of the quality of the structure. ModelAngelo has made the design decision to split chains when it is not certain about the connectivity of each chain (for example in the frequent case that loop density is relatively poor). If ModelAngelo has built an otherwise good chain, but splits it into two due to one missed loop residue, then TM-score would show the arbitrarily low score of 0.5. This is not indicative of the overall quality of the model.

In the table at the end of this rebuttal, we report TM-scores on all single-chain proteins from the CR-I-TASSER test set, where the global resolution of the experimental map extends beyond 4Å resolution. We obtained these models by building all chains in each full map simultaneously by ModelAngelo (the true problem faced by the experimentalist) but are comparing to results that the Zhang lab obtained for the easier problem of first segmenting each chain separately. The supplemental data in the CRI-I-TASSER provides information about that method's performance per chain in the experimental dataset, but because each row is not labelled with the chain ID, we could not create a scatter plot as the reviewer has done. Nevertheless, our data clearly shows that the argument of the reviewer does not hold: **ModelAngelo TM-scores** range from 0.053 to 0.998, with an **average of 0.840**, and a standard deviation of 0.216 (tmscore1 in the table below). These numbers are at odds with those provided by the reviewer.

We also ran the **TM-scores setting the denominator to be the built portion of the protein** rather than the full target (it is not clear what the reviewer used in their analysis). This provides a measure of the quality of the protein fold that is built, rather than a proxy for the completeness of each chain fragment. The values you get in this case **for ModelAngelo** are in the range of 0.348 - 0.999, with an **average of 0.946** and a standard deviation of 0.10 (tmscore2 in the table below).

Either way, according to the Supplementary Information of the **CRI-I-TASSER** paper, **average TM-scores for the method when run on the full map is 0.670**, which is lower than either of the values we observed for ModelAngelo. Although these averages admittedly also include maps with resolutions between 4-5Å, Supplementary Figure 1 and 2 in the CRI-I-TASSER paper suggests that this would not make a big difference.

I also recently found that the DeepMainmast paper currently on bioRxiv also showed a comparison with ModelAngelo, which shows similar results (<https://www.biorxiv.org/content/10.1101/2023.10.19.563151v2>) on three different datasets, including the dataset used in the ModelAngelo paper.

Another recent work, SmartFold (<https://www.biorxiv.org/content/10.1101/2023.11.02.565403v1.full.pdf>), also compares with ModelAngelo, reporting that the ModelAngelo's performance is worse than SmartFold. The SmartFold paper also compared with EMBuild, a recent publication in Nature Communications (<https://www.nature.com/articles/s41467-022-31748-9>), showing EMBuild's better performance than ModelAngelo in the single-chain dataset.

Here, the reviewer claims that ModelAngelo is no longer state-of-the-art based on two preprints that came out during the 51 days that the revised version of our paper was under review, and approximately half a year since we submitted and preprinted our original paper.

Firstly, we note that (implicitly) asking us to compare our work on these studies seems unfair, would open the possibility of endless revisions, and would set a dangerous precedent for posting preprints or publishing in conference proceedings prior to publication in Nature.

Secondly, we note that both preprints use the CR-I-TASSER test set of single-chain segmented maps, which we have argued above are not relevant to the experimentalist, and which we have shown to be suboptimal for running ModelAngelo. This will have led to unfavourable comparisons with ModelAngelo.

Thirdly, we note that the quality of both preprints is mainly derived from the docking of AlphaFold models in cryo-EM maps. This is a fundamentally different task from ModelAngelo's de-novo model building, which seeks to explain the experimental data, rather than to fit predictions to the data. Protein prediction has come a long way, and provided the prediction is correct, using predicted models to interpret cryo-EM maps may lead to better statistics. However, if one trusts protein structure prediction to begin with, then why perform the experiment? Most experiments will be performed because the predictions alone cannot be trusted. And how well will these methods perform when the prediction is incorrect? Some of these questions have recently been explored in Terwilliger et al. Nature Methods 2023 (<https://doi.org/10.1038/s41592-023-02087-4>). We envision that docking AlphaFold models into cryo-EM maps will be most useful for maps with resolutions that are too low for de-novo building approaches.

The authors compared DeepTracer, Demo-EM, Buccaneer and Phenix, which I appreciate. But except for Demo-EM, the other three methods are now baseline methods, which any new method should clearly win, thus not sufficient to claim the state-of-the-art-ness of the method. By the way, for comparing with methods, please make scatter plots, so that the values of individual maps are visible, instead of bar graphs.

In point 4 of their original review, the reviewer asked to compare ModelAngelo's performance against that of "*DeepTracer, CR-I-TASSER, DEMO-EM and Buccaneer*" for protein chains. In our revision, we included a detailed comparison with DeepTracer, DEMO-EM, Buccaneer and Phenix. Now, the reviewer calls these methods "base-line". We chose to skip CRI-I-TASSER as it was meant for single-chain atomic model building and was not suitable for comparison to these other, more general approaches. Our comparison above now shows that ModelAngelo also outperforms CR-I-TASSER.

Comparison against CryoREAD on only 3 cherry-picked maps and making conclusion out of it is totally inappropriate. I ran CryoREAD and ModelAngelo on the testset of ModelAngelo and CryoREAD datasets. The plot shown below is the sequence recall, which apparently shows substantially better performance by CryoREAD. In terms of average backbone recall, ModelAngelo had 0.842, while CryoREAD had 0.855, slightly higher than ModelAngelo. In terms of sequence recall, the gap is larger, with ModelAngelo had 0.419, while CryoREAD had 0.512. The comparison figure is attached here.

We did not cherry-pick these cases. We chose one ribosome with a low, medium and high resolution from the original test set. This was done to avoid running CryoREAD, which is very slow, on a much larger dataset.

Here, we have run ModelAngelo for the entire CryoREAD test dataset, where the resolution extends beyond 4Å. Since CryoREAD's authors have provided labeled results as a supplementary data file with their paper, in this case we could create a scatter plot to compare our results. First, we are surprised with the results the reviewer seems to be claiming for CryoREAD. These are not consistent with the numbers the authors have reported for their test set. Furthermore, there seems to be a fundamental issue with how the reviewer seems to be running ModelAngelo, getting far worse results than we have. We simply use the deposited map from the EMDB, without modification, along with the sequence file found on the PDB, and do not provide any other arguments to ModelAngelo. **The average completeness of ModelAngelo is 45.8% while that of CryoREAD is 19.6%.**

5. Page 10 and Figure 2f, “Tests, where we ran ModelAngelo without one or more of its modules, indicate that its performance comes from a combination of all three modules (Figure 2f), which is in accordance with previous observations [28].” According to Figure 2f, the order of performance of completeness (Sequence Recall) from highest to lowest is ModelAngelo > Removed IPA > Removed Sequence, but figure 3C in [28] it is Original > Seq Removed > IPA Removed. Please correct the paragraph on page 10. Based on the data provided, can one assume that the Seq and IPA modules made equal contributions?

The ablation studies in the ICLR conference proceeding and in the revised manuscript demonstrate that both the sequence and the IPA modules inject useful information into the model building process. The two papers use different test sets for these ablation studies, and the network has evolved from a beta-release of ModelAngelo to the stable release, version 1.0. The important result is that there is synergy between the three sources of information, whether one is more, or equally useful than the other is less interesting. The paragraph on page 10 reflects the results from these ablation studies adequately.

6. I mentioned that “*the authors did not provide information about the performance of amino acid type prediction. Please provide and discuss it.*” However, the author declined to reveal the results. I believe that detecting the type of amino acid accurately is informative, especially when the target sequence is unknown. Since the availability of the "model_angelo build_no_seq" command in ModelAngelo, the authors should display the accuracy of amino acid type predictions by considering the residue type with the highest probability.

This is not true. Amino acid accuracies are provided in Extended Data Table 1. These values are after the HMM postprocessing step. If the editor finds this useful, we could also add the corresponding values before this step.

7. “Similarly, what is the accuracy of base prediction for nucleic acid modeling?” This request was also neglected.

This is also not true. Base accuracies are given in Extended Data Table 2.

8. It is difficult to get a clear understanding of the overall performance of ModelAngelo for nucleotide targets based on the information presented in Figure 3. It appears that the figure only shows the successful results from the nucleotide targets. Similar to the information provided in Extended Table 1, results for the 11 ribosome structures and CRISPR-associated transpososome targets should be explained in more detail.

If the editor and the other reviewers think this would improve the paper, we could expand Extended Data Table 2 to include statistics on all examined ribosomes and the CRISPR-associated transpososome.

9. The authors totally neglected to provide data for this comment from the previous review:

“6. Also, very importantly, this version of the method should be compared with the previous version that is published in ICLR for protein structure targets. Compare the current/previous versions in terms of Calpha position prediction, RMSD, sequence level accuracy, Calpha coverage etc. We should see here clear improvement by the current method over the previous version.

Actually, we did compare the 2 versions on a dozen protein targets and surprisingly found that the current version’s performance seem to be worse than the previous method.” The author verbally replied that “Although our experience does not reflect that of the referee, we cannot guarantee that ModelAngelo 1.0 will always perform better than the beta version.” But in our analysis, the current version 1.0 is consistently worse, which questions the justification of publishing this new version in a paper:

Our manuscript describes the performance of ModelAngelo 1.0, which is the software that we distribute and that is now used by experimentalists worldwide. Whereas the beta-version of ModelAngelo could only build protein chains, version 1.0 also builds nucleotides and further modifications have been made to allow optimal detection of unknown proteins. Our paper makes no claims about the beta-version. As evidenced by the data provided in this manuscript, version 1.0 has excellent performance that will change the way structural biologists solve cryo-EM structures.

10. To this question: “There is no sequence identity check for nucleotides for building training and testing dataset. That may include some similar nucleotides structures in the testing set which has been seen in training dataset. Then the current nucleotide testing performances may be biased.”, the authors answered:

“There are much fewer nucleotide structures in the EMDB than there are protein structures. Therefore, in order to allow training of deep neural networks, it is not possible to remove all non-unique nucleotide sequences like we did for proteins. This problem will also exist for other automated model building programs.”

This is not an appropriate answer. Regardless of how other people are doing, there are ways to still address this question by showing some data. I know DNA/RNA structure modeling methods that consider sequence redundancy as much as they can.

In designing the cryoREAD test set, nucleotides with more than 80% sequence identity with nucleotides in the training set were excluded. We compared this test set with the training set of ModelAngelo. In our figure under point 4 above, we have indicated structures of the test set that do not meet this criterion for our training set in red, and structures that do meet this criterion in blue. From the plot, it follows that ModelAngelo outperforms cryoREAD regardless whether the test structure met the 80% sequence identity or not. If the editor deems it useful, we could add this figure as an additional Extended Data Figure to our manuscript.

We realise that the number of structures with less than 80% sequence homology to our training set in the CryoREAD test set is relatively low. Therefore, we also applied both cryoREAD and ModelAngelo on another 5 structures that were published in the past few months that have less than 80% sequence homology to any structure in our training set. The results are in the table below. Again, ModelAngelo is superior.

PDB	Res.	Phosphor RMSD	Backbone RMSD	Backbone recall	Backbone precision	Base accuracy	Completeness
8T2R	3.1	MA:0.68 CR: 1.14	MA:0.82 CR:2.07	MA:0.90 CR:0.75	MA:0.86 CR:0.58	MA:0.70 CR:0.51	MA:0.63 CR:0.38
8SQ9	2.9	MA:1.27 CR:1.49	MA:1.33 CR:2.32	MA:0.82 CR:0.61	MA:0.89 CR:0.72	MA:0.58 CR:0.61	MA:0.48 CR:0.37
8J12	3.1	MA:0.53 CR:1.08	MA:0.64 CR:2.00	MA:0.94 CR:0.79	MA:0.91 CR:0.71	MA:0.69 CR:0.43	MA:0.65 CR:0.34
8IBZ	3.0	MA: 1.01 CR: 1.37	MA:1.21 CR:2.20	MA:0.63 CR:0.56	MA:0.76 CR:0.45	MA:0.31 CR:0.60	MA:0.19 CR:0.33
7ZRZ	3.1	MA:0.60 CR:1.22	MA:0.73 CR:2.05	MA:0.84 CR:0.68	MA:0.89 CR:0.64	MA:0.69 CR:0.62	MA:0.58 CR:0.42

Overall, I reiterate my clear opinion that I cannot endorse further consideration of this work for publication in Nature. The authors are well respected for their work on the cryo EM program package, Relion, which I also personally use and appreciate. I would agree that having ModelAngelo in the Relion package may be useful for users. However, the performance of ModelAngelo is not state-of-the-art, with so many incomplete and inappropriate responses in this revision. The current work does not justify publication in Nature.

Above, we argue that all points raised by the reviewer are either based on flawed reasoning, contain irreproducible data presented by the reviewer, or are irrelevant to this paper. The reviewer appears to make statements from a point of expertise in cryo-EM model building and should thus have been aware of the flaws in their data and reasoning. Therefore, we can no longer exclude that this reviewer isn't acting from a potentially conflicted standpoint. As such, we do not think that we will ever be able to satisfy them. To break this impasse, we respectfully suggest that the editor consults with the other two reviewers on the points raised above, or, if deemed necessary, involves a fourth expert.

Table 1: TM-scores for ModelAngelo run on all protein chains from the CR-I-TASSER test set with resolutions beyond 4Angstroms (tm_score1 and tm_score2 use either the target or the built protein in the denominator of the TM-score).

pdb_chain	tm_score1	tm_score2
6wczB	0.0531	0.8181
6nqbE	0.1389	0.436
6nqbT	0.1531	0.5448
6tqoG	0.1632	0.7641
6spes	0.2291	0.3481
6oqva	0.2453	0.9019
3jckC	0.2655	0.9605
5m3fE	0.3508	0.9539
5x8tV	0.3626	0.7544
6spem	0.3654	0.7512
7kmbF	0.3888	0.9815
3j2pA	0.444	0.9368
6oqvX	0.4683	0.5037
6speg	0.4933	0.8895
6bvnA	0.5124	0.5412
6qulF	0.5247	0.9612
6xssA	0.5366	0.9705
6snhL	0.5367	0.9056
6spec	0.5561	0.9476
6spen	0.5679	0.9546
6tqoW	0.6451	0.8285
5nej3	0.6685	0.9747
6spel	0.701	0.9672
6tqoS	0.7111	0.9565
5nej2	0.7394	0.9846
3jckF	0.7404	0.9841
6spej	0.7535	0.8806
6spei	0.7551	0.9414
3j2pB	0.7652	0.8498
5a32A	0.7753	0.9958
6sper	0.7803	0.9443
5vmsA	0.7946	0.9739
6s6bL	0.8036	0.9021
6rqfC	0.8147	0.9814
6luII	0.8184	0.9579
5a32B	0.8184	0.9896
5x8tC	0.8244	0.9384
3jd3A	0.8281	0.9858
6j6jA	0.8307	0.9755
6rfdN	0.8347	0.9112
5wtfC	0.8631	0.9673
6g2jE	0.8643	0.9832
6sped	0.8668	0.9682
6rqfD	0.87	0.9706
6g2jV	0.876	0.9399
6spek	0.8783	0.9499

5wtfB	0.8852	0.9807
6g2jb	0.8889	0.9314
5m3fC	0.8972	0.9795
6gy6B	0.8997	0.9799
6tnna	0.9058	0.9833
6tg9D	0.9059	0.9447
6lu1M	0.9061	0.933
5m3fJ	0.9093	0.963
6tg9B	0.9143	0.9901
6rdcR	0.9178	0.9775
6s3IG	0.9198	0.9733
6spef	0.9199	0.9281
6tnnn	0.9219	0.9489
6ssep	0.9233	0.9464
3j7IA	0.9238	0.8632
6g2jW	0.9241	0.9652
6qulb	0.9278	0.9606
3j6bI	0.9296	0.9678
6g2jZ	0.9296	0.9296
3jcfA	0.93	0.9798
5vmsB	0.9303	0.9433
6speh	0.9314	0.9759
6qulU	0.9347	0.9752
6tdxI	0.9356	0.9637
3j9cA	0.9359	0.997
6s6bJ	0.9368	0.9883
2n1fA	0.9368	0.9573
6g2jA	0.9386	0.9633
6tnns	0.9407	0.97
6speq	0.9428	0.9673
6rdcS	0.9436	0.9821
6qulP	0.9438	0.9854
6speo	0.9461	0.9568
6g2jd	0.9494	0.9572
6te9F	0.9503	0.979
6d6uA	0.9506	0.9618
6qulY	0.9506	0.9885
6qulZ	0.9534	0.9688
6qulN	0.9541	0.9827
6s6bI	0.9546	0.9892
6tdxG	0.9548	0.9904
6k61L	0.9562	0.9812
6g2jq	0.9589	0.979
6qulX	0.9592	0.9846
3j6bM	0.96	0.9858
6g2jY	0.9607	0.9743
6qulS	0.9616	0.9803

6gy6A	0.9618	0.9777
6g2jX	0.9619	0.9846
6tdxH	0.962	0.9801
5uf6A	0.9636	0.9983
6g2jP	0.9643	0.9901
6oqvG	0.966	0.9903
6tg9C	0.9668	0.9865
6oqvH	0.9669	0.9811
6te9E	0.9672	0.9759
6m6cE	0.9674	0.9882
6oqvW	0.9678	0.9734
6cvbC	0.9679	0.9973
6tnne	0.968	0.9812
6wpwN	0.9685	0.9759
6tg9A	0.9696	0.9957
6qulQ	0.97	0.987
6qulC	0.9747	0.9967
6qulW	0.9752	0.9855
6qula	0.9757	0.9757
6smqB	0.9761	0.9645
6tnnX	0.9773	0.9916
6s3IF	0.9776	0.9933
6vykA	0.9784	0.9858
6rqfA	0.9788	0.9879
6rqfB	0.979	0.9913
6qulR	0.9793	0.9962
6s6bH	0.9797	0.986
6tnnY	0.9802	0.985
6oikB	0.9806	0.9922
6smlA	0.9819	0.9963
4ci0B	0.982	0.982
6tnng	0.9825	0.9907
6g2jN	0.9839	0.9896
6qulD	0.9845	0.9939
6k61E	0.9845	0.9845
4ci0C	0.985	0.9921
6rfdB	0.9851	0.9943
6k61F	0.9872	0.9942
3jciA	0.9883	0.9935
6qulK	0.9883	0.9953
5mpp0	0.9901	0.9901
3j9sA	0.9914	0.9964
4ci0A	0.993	0.9956
6g2jM	0.9937	0.9959
6cvbB	0.9943	0.9983
6k61B	0.9979	0.9992